# Characterization of cancer-driving nucleotides (CDNs) across genes, cancer types, and patients

Lingjie Zhang[1], Tong Deng[1], Zhongqi Liufu[1,2], Xiangnyu Chen[1], Shijie Wu[1], Xueyu Liu[1], Changhao Shi[1], Bingjie Chen[1,3], Zheng Hu[4], Qichun Cai[5], Chenli Liu[4], Mengfeng Li[6], Miles E Tracy[1], Xuemei Lu[2], Chung-I Wu[1,7]*, Hai-Jun Wen[1]*

[1]State Key Laboratory of Biocontrol, School of Life Sciences, Sun Yat-sen University, Guangzhou, China; [2]Center for Excellence in Animal Evolution and Genetics, The Chinese Academy of Sciences, Kunming, China; [3]GMU-GIBH Joint School of Life Sciences, Guangzhou Medical University, Guangzhou, China; [4]CAS Key Laboratory of Quantitative Engineering Biology, Shenzhen Institute of Synthetic Biology, Shenzhen Institute of Advanced Technology, Chinese Academy of Sciences, Shenzhen, China; [5]Cancer Center, Clifford Hospital, Jinan University, Guangzhou, China; [6]Cancer Research Institute, School of Basic Medical Sciences, Southern Medical University, Guangzhou, China; [7]Department of Ecology and Evolution, University of Chicago, Chicago, United States

*For correspondence:
ciwu@uchicago.edu (C-IW);
wenhj5@mail.sysu.edu.cn (H-JW)

Competing interest: The authors declare that no competing interests exist.

## eLife Assessment

This **valuable** study is a companion to a paper introducing a theoretical framework and methodology for identifying cancer-driving nucleotides (CDNs). The evidence that recurrent SNVs or CDNs are common in true cancer driver genes is **convincing**, with more limited evidence that many more undiscovered cancer driver mutations will have CDNs, and that this approach could identify these undiscovered driver genes with about 100,000 samples.

**Abstract** A central goal of cancer genomics is to identify, in each patient, all the cancer-driving mutations. Among them, point mutations are referred to as cancer-driving nucleotides (CDNs), which recur in cancers. The companion study shows that the probability of $i$ recurrent hits in **n** patients would decrease exponentially with $i$; hence, any mutation with $i \geq 3$ hits in The Cancer Genome Atlas (TCGA) database is a high-probability CDN. This study characterizes the 50–150 CDNs identifiable for each cancer type of TCGA (while anticipating 10 times more undiscovered ones) as follows: (i) CDNs tend to code for amino acids of divergent chemical properties. (ii) At the genic level, far more CDNs (more than fivefold) fall on noncanonical than canonical cancer-driving genes (CDGs). Most undiscovered CDNs are expected to be on unknown CDGs. (iii) CDNs tend to be more widely shared among cancer types than canonical CDGs, mainly because of the higher resolution at the nucleotide than the whole-gene level. (iv) Most important, among the 50–100 coding region mutations carried by a cancer patient, 5–8 CDNs are expected but only 0–2 CDNs have been identified at present. This low level of identification has hampered functional test and gene-targeted therapy. We show that, by expanding the sample size to $10^5$, most CDNs can be identified. Full CDN identification will then facilitate the design of patient-specific targeting against multiple CDN-harboring genes.

## Introduction

Tumorigenesis in each patient is driven by mutations in the patient's genome. Hence, a central goal of cancer genomics is to identify *all* driving mutations in each patient. This task is particularly challenging because each driving mutation is present in only a small fraction of patients. As the number of driver mutations in each patient has been estimated to be >5 (*Armitage and Doll, 1954*; *Bozic et al., 2010*; *Hanahan and Weinberg, 2011*; *Belikov, 2017*; *Anandakrishnan et al., 2019*), the total number of driver mutations summed over all patients must be quite high.

This study, together with the companion paper (*Zhang et al., 2024*), is based on one simple premise: in the massively repeated evolution of cancers, any advantageous cancer-driving mutation should recur frequently, say, $i$ times in **n** patients. The converse that nonrecurrent mutations are not advantageous is part of the same premise. We focus on point mutations, referred to as cancer-driving nucleotides (CDNs), and formulate the maximum of $i$ (denoted $i^*$) in **n** patients if mutations are not advantageous. For example, in The Cancer Genome Atlas (TCGA) database with **n** generally in the range of 500–1000, $i^* = 3$. Hence, any point mutation with $i \geq 3$ is a CDN. At present, a CDN would have a prevalence of 0.3% among cancer patients. If the sample size approaches $10^6$, a CDN only needs to be prevalent at $5 \times 10^{-5}$, the theoretical limit (*Zhang et al., 2024*).

Although there are many other driver mutations (e.g., fusion genes, chromosomal aberrations, epigenetic changes, etc.), CDNs should be sufficiently numerous and quantifiable to lead to innovations in functional tests and treatment strategies. Given the current sample sizes of various databases (*Cerami et al., 2012*; *Weinstein et al., 2013*; *Tate et al., 2019*; *de Bruijn et al., 2023*), each cancer type has yielded 50–150 CDNs while the CDNs to be discovered should be at least 10 times more numerous. The number of CDNs currently observed in each patient is 0–2 for most cancer types. This low level of discovery has limited functional studies and hampered treatment strategies.

While we are proposing the scale-up of sample size to discover most CDNs, we now characterize CDNs that have been discovered. The main issues are the distributions of CDNs among genes, across cancer types, and, most important, among patients. In this context, cancer driver genes (CDGs) would be a generic term. We shall use 'canonical CDGs' (or conventional CDGs) for the driver genes in the union set of three commonly used lists (*Bailey et al., 2018*; *Sondka et al., 2018*; *Martínez-Jiménez et al., 2020*). In parallel, CDN-harboring genes, referred to as 'CDN genes', constitute a new and expanded class of CDGs.

The first issue is that CDNs are not evenly distributed among genes. The canonical cancer drivers such as *TP53*, *KRAS*, and *EGFR* tend to have many CDNs. However, the majority of CDNs, especially those yet-to-be-identified ones, may be rather evenly distributed with each gene harboring only 1–2 CDNs. Hence, the number of genes with tumorigenic potential may be far larger than realized so far. The second issue is the distribution of CDNs and CDGs among cancer types. It is generally understood that the canonical CDGs are not widely shared among cancer types. However, much (but not all) of the presumed cancer-type specificity may be due to low statistical resolution at the genic level.

The third issue concerns the distribution of CDNs among patients. Clearly, the CDN load of a patient is crucial in diagnosis and treatment. However, the conventional diagnosis at the gene level may have two potential problems. One is that many CDNs do not fall in canonical CDGs as signals from one or two CDNs get diluted. Second, a canonical CDG, when mutated, may be mutated at a non-CDN site. In those patients, the said CDG does not drive tumorigenesis. We shall clarify the relationships between CDN mutations and genes that may or may not harbor them.

The characterizations of discovered CDNs are informative and offer a road map for expanding the CDN list. A complete CDN list for each cancer type will be most useful in functional test, diagnosis, and treatment. A full list of mutations that drive the evolution of complex traits is at the center of evolutionary genetics. Such phenomena as complex human diseases (e.g., diabetes) (*Vujkovic et al., 2020*; *Lagou et al., 2023*; *Xue et al., 2023*; *Suzuki et al., 2024*), the genetics of speciation (*Chen et al., 2022b*; *Wang et al., 2022*; *Wu, 2022*), and the evolution of viruses in epidemics (*Deng et al., 2022*; *Ruan et al., 2022*; *Cao et al., 2023*; *Ruan et al., 2023*) are all prime examples in need of a full list. Thanks to their massively repeated evolution, cancers could be the first complex systems well resolved at the genic level.

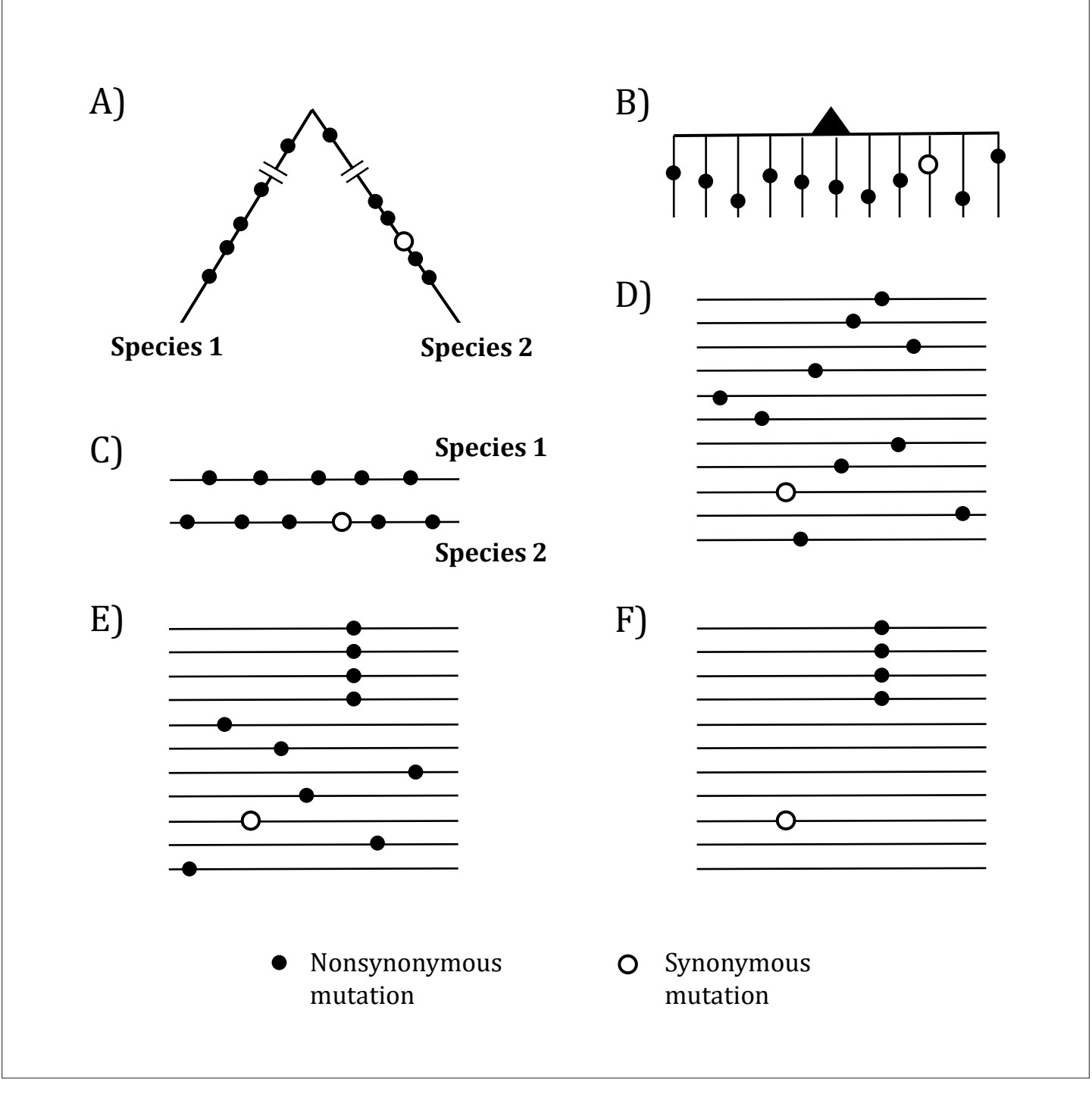

**Figure 1.** Mutations in organismal evolution vs. cancer evolution. (**A, B**) A hypothetical example of DNA sequence evolution in organism vs. in cancer with the same number of mutations. (**C**) Mutation distribution in two species in the organismal evolution of (**A**). (**D, E**) Mutation distribution in cancer evolution among 10 sequences may have D and E patterns. (**F**) Another pattern of mutation distribution in cancer evolution with a recurrent site but shows too few total mutations. Mutations of (**F**) are cancer-driving nucleotides (CDNs) missed in the conventional screens.

## Results

In molecular evolution, a gene under positive selection is recognized by its elevated evolutionary rate (*Figure 1A and C*). There have been numerous methods for determining the extent of rate elevation (*Li et al., 1985*; *Nei and Gojobori, 1986*; *Yang and Swanson, 2002*; *Lawrence et al., 2013*;

*Martincorena et al., 2017*; *Pan et al., 2022*; *Sherman et al., 2022*; *Wang et al., 2022*; *Ruan et al., 2023*), and cancer evolution studies have adopted many of them. However, no model has been developed to take advantage of the massively repeated evolution of cancers (*Figure 1B*), which happens in tens of millions of people at any time.

In the whole-gene analysis, *Figure 1C–E* are identical, each with $A:S$ = 10:1, where $A$ and $S$ denote nonsynonymous and synonymous mutations, respectively. However, the presence of a four-hit site in *Figure 1E* is far less likely to be neutral than *Figure 1C and D*. Although the ratio in *Figure 1F*, $A:S$ = 4:1, is statistically indistinguishable from the neutral ratio of about 2.5:1, *Figure 1F* in fact has much more power to reject the neutral ratio than *Figure 1C and D*. After all, the probability that multiple hits are at the same site in a big genome is obviously very small.

## The analyses of CDNs across the whole genome

For the entire coding regions in the cancer genome data, we define $A_i$ (or $S_i$) as the number of nonsynonymous (or synonymous) sites that harbor a mutation with $i$ recurrences. *Table 1* presents the distribution of $A_i$ and $S_i$ across the 12 cancer types with **n** > 300 (*Weinstein et al., 2013*).

For neutral mutations, we define $i^*$ as the threshold above which the expected numbers of $A_i$ would be <1, that is, $E\left[A_{i \geq i*}\right] < 1$, The corollary is that all $A_{i \geq i*}$ sites are advantageous CDNs. (Since $S_i$ is ~$A_i$/2.3, the same $i^*$ would apply to $S_i$ as well: $E\left[S_{i \geq i*}\right] < 1$.) As $i^*$ is a function of the number of patients (**n**), it is shown mathematically in the companion study (*Zhang et al., 2024*) that $i^*$ = 3 for **n** < 1000. Interestingly, while the $E\left[A_{i \geq 3}\right]$ is < 1, the expected $E\left[A_{i \geq 4}\right]$ is ≪ 1, in the order of 0.001. Hence, $i^*$ = 4 may be considered unnecessarily stringent.

We should note that this study is constrained by **n** < 1000 in TCGA databases. (Databases with larger **n**s are also used where the actual **n**s are often uncertain.) At $i^*$ = 3, we could detect only a fraction (<10%; see below) of CDNs. Many more tumorigenic mutations may be found in the $i$ = 1 or 2 classes although not every one of them is a CDN. Since these two classes of mutations are far more numerous, they should account for the bulk of CDNs to be discovered. Indeed, *Table 1* shows 76 $A_{i \geq 3}$ CDN mutations per cancer type but 681 $A_2$ and 56,648 $A_1$ mutations in the lower recurrence groups. If **n** reaches $10^{5-6}$, most of the undiscovered CDNs in the $A_1$ and $A_2$ classes should be identified (*Zhang et al., 2024*).

In *Table 2*, we estimate the proportion of the $A_1$ and $A_2$ mutations that are possible CDNs. The relationships of $A_3/S_3 > A_2/S_2$, $A_2/S_2 > A_1/S_1$, and $A_1/S_1 > A_0/S_0$ are almost always observed in *Table 1* with 32 (3 × 8 + 2 × 4) out of 36 such relationships. The use of $A/S$ ratios may still underestimate the selective advantages of $A_{1-3}$ mutations because $S_{1-3}$ may have slight advantages as well (*Zhang et al., 2024*). Assuming $S_1$ is truly neutral, we use $S_0$ to $S_1$ as the basis to calculate the excess of $A_{1-3}$ in *Table 2* where 35 of the 36 $Obs(A_i) > Exp(A_i)$ relationships can be observed. The implication is that hundreds and, likely low thousands, of $A_1$s and $A_2$' should be CDNs, whereas we have only confidently identified ~76 strong CDNs, on average, for a cancer type. (Note that $A_1$ excesses are less reliable since a 1% error in the calculation would mean 566 CDNs.)

## CDNs and the amino acids affected

We now ask whether the amino acid changes associated with CDNs bear the signatures of positive selection. Amino acids that have divergent physico-chemical properties have been shown to be under strong selection, both positive and negative (*Chen et al., 2019a*; *Chen et al., 2019b*; *Chen et al., 2022b*). We note that, in almost all cases in cancer evolution, when a codon is altered, only one nucleotide of the triplet codon is changed. Among the 190 amino acid (AA, 20×19/2) pairs, only 75 of the pairs differ by 1 bp (*Tang et al., 2004*). For example, Pro (CCN) and Ala (GCN) may differ by only 1 bp but Pro and Gly (GGN) must differ by at least 2 bp. These 75 AA changes, referred to as the elementary AA changes (*Grantham, 1974*; *Li et al., 1985*; *Yang et al., 2003*; *Meyer et al., 2021*), account for almost all AA substitutions in somatic evolution.

In a series of studies (*Tang et al., 2004*; *Chen et al., 2019a*; *Chen et al., 2019b*), we have defined the physico-chemical distances between AAs of the 75 elementary pairs as $\Delta U_i$, where $i$ = 1–75. $\Delta U_i$ reflects 47 measures of AA differences including hydrophobicity, size, charge, etc., and ranges between 0 and 1. The most similar pair, Ser and Thr, has $\Delta U_i$ = 0, and the most dissimilar pair is Asp and Try with $\Delta U_i$ = 1. These studies show that $\Delta U_i$ is a strong determinant of the evolutionary rates of DNA sequences and that large-step changes (i.e., large $\Delta U_i$ s) are more acutely 'recognized'

**Table 1.** Mutation recurrences ($A_i$'s and $S_i$'s) in 12 cancer types.

| | Lung | Breast | Central nervous system | Kidney | Upper aerodigestive tract | Colon | Endometrium | Prostate | Stomach | Urinary tract | Ovary | Liver | Average |
|---|---|---|---|---|---|---|---|---|---|---|---|---|---|
| Patients # | 1035 | 963 | 873 | 711 | 688 | 571 | 465 | 465 | 423 | 404 | 404 | 367 | 614 |
| *$A_0$ | 22,540,623 | 21,683,136 | 20,783,835 | 22,247,653 | 21,580,444 | 20,601,026 | 20,766,001 | 21,300,810 | 20,892,755 | 21628918 | 22278124 | 22618059 | 21576782 |
| *$S_0$ | 78,042,81 | 9,388,418 | 10,298,911 | 87,814,83 | 93,332,83 | 10,428,913 | 10,375,596 | 97,543,31 | 10,243,634 | 9426888 | 8746002 | 8255268 | 9403084 |
| $A/S\_0$ | 2.89 | 2.31 | 2.02 | 2.53 | 2.31 | 1.98 | 2.00 | 2.18 | 2.04 | 2.29 | 2.55 | 2.74 | 2.29 |
| $A_1$ | 195958 | 44696 | 25122 | 25669 | 66924 | 94634 | 78870 | 9583 | 78834 | 66153 | 21138 | 25731 | 61109 |
| $S_1$ | 69393 | 16732 | 10182 | 9317 | 26151 | 38606 | 31982 | 3613 | 32538 | 26546 | 7227 | 9398 | 23474 |
| $A/S\_1$ | 2.82 | 2.67 | 2.47 | 2.76 | 2.56 | 2.45 | 2.47 | 2.65 | 2.42 | 2.49 | 2.92 | 2.74 | 2.60 |
| $A_2$ | 2946 | 233 | 287 | 56 | 489 | 1662 | 1052 | 29 | 1176 | 816 | 51 | 46 | 737 |
| $S_2$ | 969 | 62 | 75 | 11 | 159 | 736 | 386 | 9 | 489 | 308 | 9 | 12 | 249 |
| $A/S\_2$ | 3.04 | 3.76 | 3.83 | 5.09 | 3.08 | 2.26 | 2.73 | 3.22 | 2.40 | 2.65 | 5.67 | 3.83 | 2.74 |
| $A_3$ | 99 | 18 | 42 | 14 | 28 | 91 | 52 | 6 | 79 | 60 | 9 | 9 | 42.3 |
| $S_3$ | 21 | 2 | 6 | 1 | 5 | 28 | 11 | 0 | 14 | 9 | 0 | 0 | 8.08 |
| $A/S\_3$ | 4.71 | 9 | 7 | 14 | 5.6 | 3.25 | 4.73 | 6.0 | 5.64 | 6.67 | 9.0 | 9.0 | 5.23 |
| †$A_{i \geq 3}$ | 178 | 51 | 84 | 18 | 77 | 148 | 142 | 14 | 124 | 100 | 26 | 23 | 82.1 |
| †$A_{i \geq 4}$ | 79 | 33 | 42 | 4 | 49 | 57 | 90 | 8 | 45 | 40 | 17 | 14 | 39.8 |
| $A_4$ | 23 | 10 | 8 | 2 | 14 | 23 | 21 | 3 | 23 | 11 | 4 | 3 | 11.1 |
| $A_5$ | 16 | 6 | 10 | 2 | 10 | 6 | 20 | 2 | 9 | 9 | 3 | 5 | 8.2 |
| $A_{6-9}$ | 27 | 10 | 10 | 0 | 13 | 9 | 32 | 2 | 7 | 12 | 6 | 2 | 10.8 |
| $A_{[10, 20)}$ | 7 | 3 | 10 | 0 | 9 | 11 | 9 | 1 | 6 | 5 | 4 | 4 | 5.75 |
| $A_{\geq 20}$ | 6 | 4 | 4 | 0 | 3 | 8 | 8 | 0 | 0 | 3 | 0 | 0 | 3 |
| ‡Total | 202828 | 45669 | 26596 | 25841 | 68387 | 98931 | 81898 | 9706 | 81678 | 68297 | 21387 | 25944 | 63097 |
| SiteNbr | 22739705 | 21728116 | 20809328 | 22273396 | 21647934 | 20697470 | 20846065 | 21310436 | 20972889 | 21695987 | 22299339 | 22643859 | 21638710 |
| nE(u) | 9.07E-03 | 1.79E-03 | 1.00E-03 | 1.06E-03 | 2.83E-03 | 3.84E-03 | 3.15E-03 | 3.72E-04 | 3.27E-03 | 2.88E-03 | 8.28E-04 | 1.14E-03 | 2.6E-03 |

*See 'Methods' for the calculations of $A_0$ and $S_0$.

†$A_i$ and $S_i$ are as defined in the text.

‡'Total' represents the total number of missense mutations, or . 'Site number' refers to the count of missense sites. nE(u) is calculated based on synonymous mutations, representing the expected number of neutral mutations per site in a population of size n.

**Table 2.** Excess of $A_i$s of each $i$ class.

| Recurrences | Lung | Breast | Central nervous system | Kidney | Upper aerodigestive tract | Colon | Endometrium | Prostate | Stomach | Urinary tract | Ovary | Liver |
|---|---|---|---|---|---|---|---|---|---|---|---|---|
| *$A_{1\_o}$ | 195958 | 44696 | 25122 | 25669 | 66924 | 94634 | 78870 | 9583 | 78834 | 66153 | 21138 | 25731 |
| *,†$A_{1\_e}$ | 198627 | 38586 | 20532 | 23582 | 60316 | 76049 | 63860 | 7888 | 66194 | 60751 | 18396 | 25720 |
| Excess | −2669 | 6110 | 4590 | 2087 | 6608 | 18585 | 15010 | 1695 | 12640 | 5402 | 2742 | 11 |
| ‡Ratio (%) | −1.36 | 13.67 | 18.27 | 8.13 | 9.87 | 19.64 | 19.03 | 17.69 | 16.03 | 8.17 | 12.97 | 0.04 |
| $A_{2\_o}$ | 2946 | 233 | 287 | 56 | 489 | 1662 | 1052 | 29 | 1176 | 816 | 51 | 46 |
| $A_{2\_e}$ | 1750 | 69 | 20 | 25 | 169 | 280 | 196 | 3 | 210 | 171 | 15 | 29 |
| Excess | 1195.61 | 164.36 | 266.72 | 31.01 | 320.48 | 1381.54 | 855.77 | 26.08 | 966.42 | 645.41 | 35.81 | 16.75 |
| Ratio (%) | 40.58 | 70.54 | 92.93 | 55.37 | 65.54 | 83.13 | 81.35 | 89.93 | 82.18 | 79.09 | 70.22 | 36.42 |
| $A_{3\_o}$ | 99 | 18 | 42 | 14 | 28 | 91 | 52 | 6 | 79 | 60 | 9 | 9 |
| $A_{3\_e}$ | 15.43 | 0.12 | 0.02 | 0.03 | 0.47 | 1.03 | 0.60 | 0.00 | 0.66 | 0.48 | 0.01 | 0.03 |
| Excess | 83.57 | 17.88 | 41.98 | 13.97 | 27.53 | 89.97 | 51.40 | 6.00 | 78.34 | 59.52 | 8.99 | 8.97 |
| Ratio (%) | 84.42 | 99.32 | 99.95 | 99.81 | 98.32 | 98.86 | 98.84 | 99.98 | 99.16 | 99.20 | 99.86 | 99.63 |
| $A_{4\_o}$ | 23 | 10 | 8 | 2 | 14 | 23 | 21 | 3 | 23 | 11 | 4 | 3 |
| $A_{4\_e}$ | 0.13593 | 0.00022 | 1.98E-05 | 2.81E-05 | 0.00132 | 0.00381 | 0.00185 | 4.00E-07 | 0.00210 | 0.00135 | 1.04E-05 | 3.78E-05 |
| Excess | 22.8641 | 9.99978 | 7.99998 | 1.99997 | 13.9987 | 22.9962 | 20.9981 | 3 | 22.9979 | 10.9987 | 3.99999 | 2.99999 |
| Ratio (%) | 99.41 | 100 | 100 | 100 | 99.99 | 99.98 | 99.99 | 100.00 | 99.99 | 99.99 | 100 | 100.00 |

*The notation of 'o' and 'e' following $A_i$s represents the observed $A_i$ and expected $A_i$.

†See 'Methods' for the calculation of expected $A_i$'s.

‡Ratio is the proportion of observed sites in excess, that is, the proportion of putative CDNs in the observation.

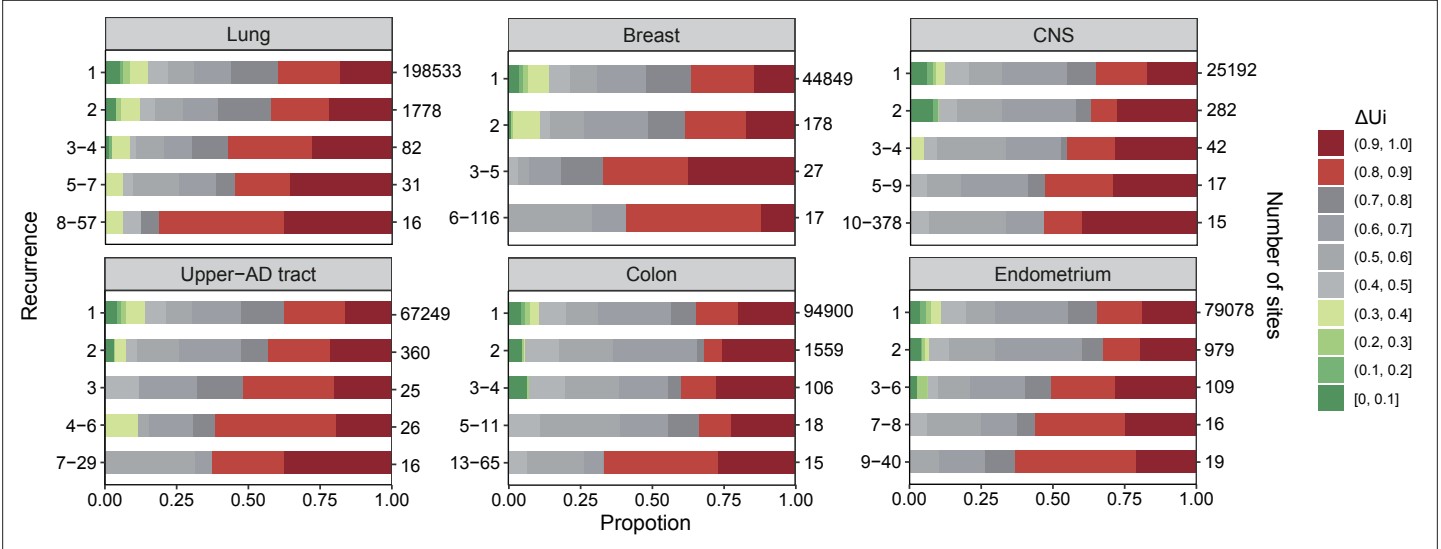

**Figure 2.** $\Delta Ui$ analysis across six cancer types. $\Delta Ui$, ranging between 0 and 1 (*Tang et al., 2004*; *Chen et al., 2019a*), is a measure of physico-chemical differences among the 20 amino acids (see the text). The most similar amino acids have $\Delta Ui$ near 0 and the most dissimilar ones have $\Delta Ui$ near 1. Each panel corresponds to one cancer type, with horizontal bar represents $\Delta Ui$ distribution of each recurrence group. The numbers on the left of the panel are *i* values and on the right are the number of sites. Note that the proportion of dark red segments increases as *i* increases. This figure shows that mutations at high recurrence sites (larger *i*s) code for amino acids that are chemically very different from the wild type.

by natural selection. These large-step changes are either highly deleterious or highly advantageous. Most strikingly, advantageous mutations are enriched with AA pairs of $\Delta Ui > 0.8$ (*Chen et al., 2019a*).

To analyze the properties of CDNs, we choose six cancer types from *Table 1* that have the largest sample sizes (**n** > 500) but leap over kidney since kidney cancers have unusually low CDN counts. In *Figure 2*, we divide the CDNs into groups according to the number of recurrences, *i*. CDNs of similar *i*s are merged into the same group in the descending order of *i*, until there are at least 10 CDNs in the group. The six cancer types show two clear trends: (1) the proportion of CDNs with $\Delta Ui > 0.8$ (red color segments) increases in groups with higher recurrences; and (2) in contrast, the proportion of CDNs with $\Delta Ui < 0.4$ (green segments) decreases as recurrences increase. These two trends would mean that highly recurrent CDNs tend to involve larger AA distances ($\Delta Ui > 0.8$) and similar AAs tend not to manifest strong fitness increases. In general, CDNs alter amino acids in ways that expose the changes to strong selection.

## CDNs in relation to the genes harboring them

We shall use the term 'CDN genes' for genes having at least one CDN site. Since CDN genes contribute to tumorigenesis when harboring a CDN mutation, they should be considered cancer drivers as well. CDN genes have two desirable qualities for recognition as driver genes. First, CDNs are straight-forward and unambiguous to define (e.g., $i \geq 3$ for **n** < 1000). In the literature, there have been multiple definitions of CDGs (*Reimand and Bader, 2013*; *Porta-Pardo and Godzik, 2014*; *Mularoni et al., 2016*; *Arnedo-Pac et al., 2019*), resulting in only modest overlaps among cancer gene lists (*Appendix 1—figure 1*). Second, the evolutionary fitness of CDN, and hence the tumorigenic potentials of CDN genes, can be computed (Appendix 2, section 'Quantifying evolutionary fitness of CDN').

We now present the analyses of CDN genes using the same six cancers of *Figure 2*. Two types of CDN genes are shown in *Table 3*. Type I genes fulfill the conventional criterion of fast evolution with the whole-gene Ka/Ks (or dN/dS) significantly larger than 1 (*Martincorena et al., 2017*). Averaged across cancer types, type I overlaps by 95.7% with the canonical CDG list, which is the union of three popular lists (*Bailey et al., 2018*; *Sondka et al., 2018*; *Martínez-Jiménez et al., 2020*). Type I genes are mostly well-known canonical CDGs (e.g., *TP53*, *PIK3CA*, and *EGFR*).

Type II (CDN genes) is the new class of CDGs. These genes have CDNs but do not meet the conventional criteria of whole-gene analysis. Obviously, if a gene has only one or two CDNs plus some sporadic hits, the whole-gene Ka/Ks would not be significantly greater than 1. As shown in *Table 3*,

**Table 3.** Distribution of cancer-driving nucleotides (CDNs) among genes.

| CDN calls based on i*=3 | Lung | Breast | Central nervous system | Upper aerodigestive tract | Colon | Endometrium | Mean | †Total | Overlap with the conventional set | Criteria of classification |
|---|---|---|---|---|---|---|---|---|---|---|
| # of patients (n) | 1035 | 963 | 873 | 688 | 571 | 465 | - | - | - | |
| CDN count | 178 | 50 | 83 | 77 | 148 | 142 | 113.3 | 495 | - | |
| **# CDN-carrying genes (type I fulfills the convention of ‡Ka/Ks > 1**; type II does not)** | | | | | | | | | | |
| Type I (Ka/Ks >1") | 10 | 8 | 12 | 13 | 10 | 21 | 12.33 | 45 | 95.7% | Conventional |
| Type II (Ka/Ks ~ 1) | 79 | 9 | 12 | 19 | 86 | 35 | 40 | 229 | 26.1% | This study only |
| All CDN genes | 89 | 17 | 24 | 32 | 96 | 56 | 52.33 | 258 | 47% | Both types |
| Genes with 1–2 CDNs (% all CDN genes) | 80 (89.9 %) | 14 (82.4 %) | 19 (79.2 %) | 27 (84.4 %) | 90 (93.8 %) | 45 (80.4 %) | 45.8 (85 %) | 250 (96.9 %) | | A subset of both types |
| **Number of driver genes in three major CDG lists** | | | | | | | | | | |
| *Other criteria: | | | | | | | | | – | Variable (see legends) |
| IntOGen | 118 | 100 | 100 | 106 | 86 | 72 | 97 | 321 | | |
| Bailey et al. | 36 | 29 | 32 | 38 | 20 | 55 | 35 | 134 | | |
| CGC Tier 1 | 30 | 32 | 32 | 24 | 44 | 23 | 30.83 | 118 | | |

*IntOGen, Bailey et al., and CGC Tier 1 are the three major CDG lists adopted here for comparison (*Bailey et al., 2018*; *Sondka et al., 2018*; *Martínez-Jiménez et al., 2020*).

†"Total" refers to the cumulative number of unique genes identified across all six cancer types.

‡Here, ** denotes significant Ka/Ks results with a corrected q-value < 0.1 based on dndscv analysis.

over 80% of CDN genes have only 1–2 CDN sites. The salient result is that type II genes outnumber type I genes by a ratio of 5:1 (229:45, column 8, *Table 3*). Furthermore, type II genes overlap with the canonical CDG list by only 23%.

Type II genes represent a new class of cancer drivers that concentrate their tumorigenic strength on a small number CDN sites. They have been missed by the conventional whole-gene definition of cancer drivers. One such example is the *FGFR3* gene in lung cancer. This gene of 809 codons has only eight hits, among which one is a CDN ($i = 3$) in lung cancer. It is noticed solely for this CDN. In Appendix 2, section 'Functional annotation of new cancer drivers', we briefly annotate these new CDGs for comparisons with the canonical driver genes. Possible functional tests in the future can be found in 'Discussion'.

We now briefly discuss the driver genes listed in previous studies as shown at the lower part of *Table 3* (*Bailey et al., 2018*; *Sondka et al., 2018*; *Martínez-Jiménez et al., 2020*). From the total number of CDGs listed, it is clear that the overlaps are limited. As analyzed before (*Wu et al., 2016*), conventional gene lists overlap mainly by a core set of high Ka/Ks genes. This core set has not changed much as various criteria such as the replication timing, expression profiles, and epigenetic features are introduced. These criteria are the reasons for the many CDGs recognized by only a small subset of CDG lists. CDN genes, in contrast, can be objectively defined as CDN mutations ($i$ recurrences in **n** samples) themselves are unambiguous.

## Variation in CDN number and tumorigenic contribution among genes

By and large, the distribution of CDNs among genes is very uneven. *Figure 3A* shows 10 genes with at least six CDNs, whereas 87 genes have only one CDN. Two genes stand out for the number of CDNs they harbor, *TP53* and *PIK3CA*, which also happen to be the only genes mutated in >15% of all cancer patients surveyed (*Kandoth et al., 2013*). Clearly, the prevalence of mutations in a gene is a function of the number of strong CDNs it harbors.

Although a small number of genes have unusually high number of CDNs, these genes may not drive the tumorigenesis in proportion to their CDN numbers in individual patients. *Figure 3B* shows the number of CDN mutations on *TP53* that occur in any single patient. Usually, only one CDN change is observed in a patient, whereas two or three CDN mutations are expected. It thus appears that CDNs on the same genes are redundant in their tumorigenic effects such that the second hit may not yield additional advantages. This pattern of disproportionally lower contribution by CDN-rich genes is true in other genes such as *EGFR* and *KRAS*. Consequently, the large number of genes with only one or two CDN sites are disproportionately important in driving the tumorigenesis of individual patients.

## CDNs in relation to the cancer types: The pan-cancer properties

In the current literature, CDGs (however they are defined) generally meet the statistical criteria for driver genes in only one or a few cancer types. However, genes may in fact contribute to tumorigenesis but are insufficiently prevalent to meet the statistical requirements for CDGs. Many genes are indeed marginally qualified as drivers in some tissues and barely miss the statistical cutoff in others. To see if genes that drive tumorigenesis in multiple tissues are more common than currently understood, we need to raise the sensitivity of cancer driver detection. Thus, CDNs may provide the resolution.

To test the pan-cancer-driving capacity of CDNs, we define $i_{max}$ as the largest $i$ values among the 12 cancer types for each CDN. The number of cancer types where the said mutation can be detected (i.e., $i > 0$) is designated NC12. *Figure 4* presents the relationship between the observed NC12 of each CDN against $i_{max}$ of that CDN. Clearly, many CDNs are observed in multiple cancer types (NC12 > 3), even though they do not qualify as a driver gene in all but a single cancer type. It happens frequently when a mutation has $i > 3$ in one cancer type but has $i < 3$ in others. One extreme example is C394 and G395 in *IDH1*. In central nervous system (CNS), both sites show $i \gg 3$, while in six other cancer types (lung, breast, large intestine, prostate, urinary tract, liver), their hits are $i < 3$ but > 0. Conditional on a specific site informed by a cancer type, a mutation in another cancer type should be very unlikely if the mutation is not tumorigenic in multiple tissues. Hence, the pattern in *Figure 4* is interpreted to be drivers in multiple cancer types, but with varying statistical strength.

Examining *Figure 4* more carefully, we could see that CDNs with a larger $i_{max}$ in one cancer type are more likely to be identified as CDNs in multiple cancer types (red dots, $r = 0.97$, p=$9.23 \times 10^{-5}$, Pearson's correlation test). Of 22 sites with $i_{max} > 20$, 15 are identified as CDNs ($i \geq 3$) in multiple cancer

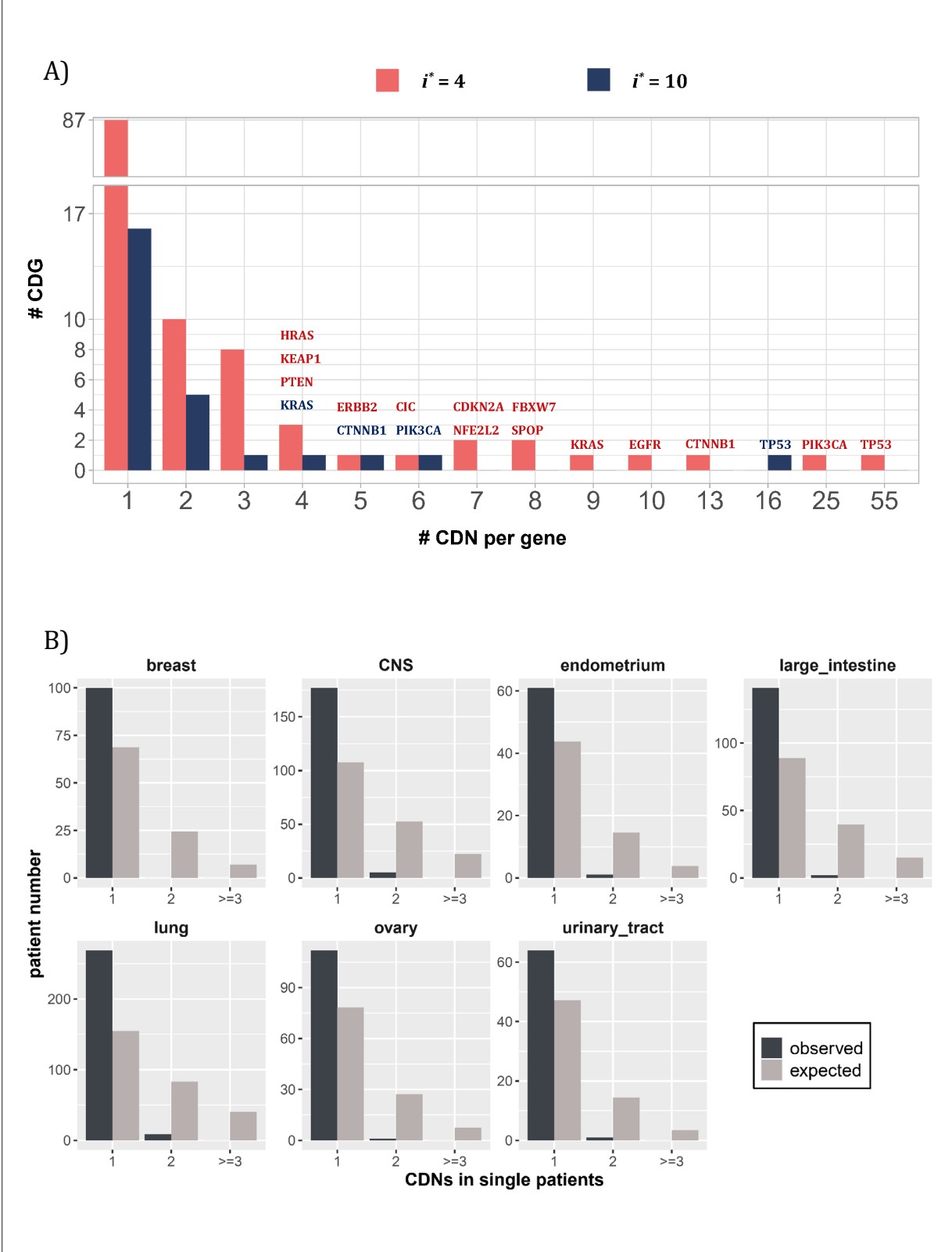

**Figure 3.** Distribution of cancer-driving nucleotides (CDNs) among genes. (**A**) Out of 119 CDN-carrying genes (red bars), 87 have only one CDN. For the rest, *TP53* possesses the most CDNs with three others having more than 10 CDNs. (**B**) CDN number in *TP53* among patients. The dark bar represents the observed patient number with corresponding CDNs of the X-axis. The gray bar shows the expected patient distribution. Clearly, *TP53* only needs to contribute one CDN to drive tumorigenesis. Hence, *TP53* (and other canonical driver genes; see text), while prevalent, does not contribute disproportionately to the tumorigenesis of each patient.

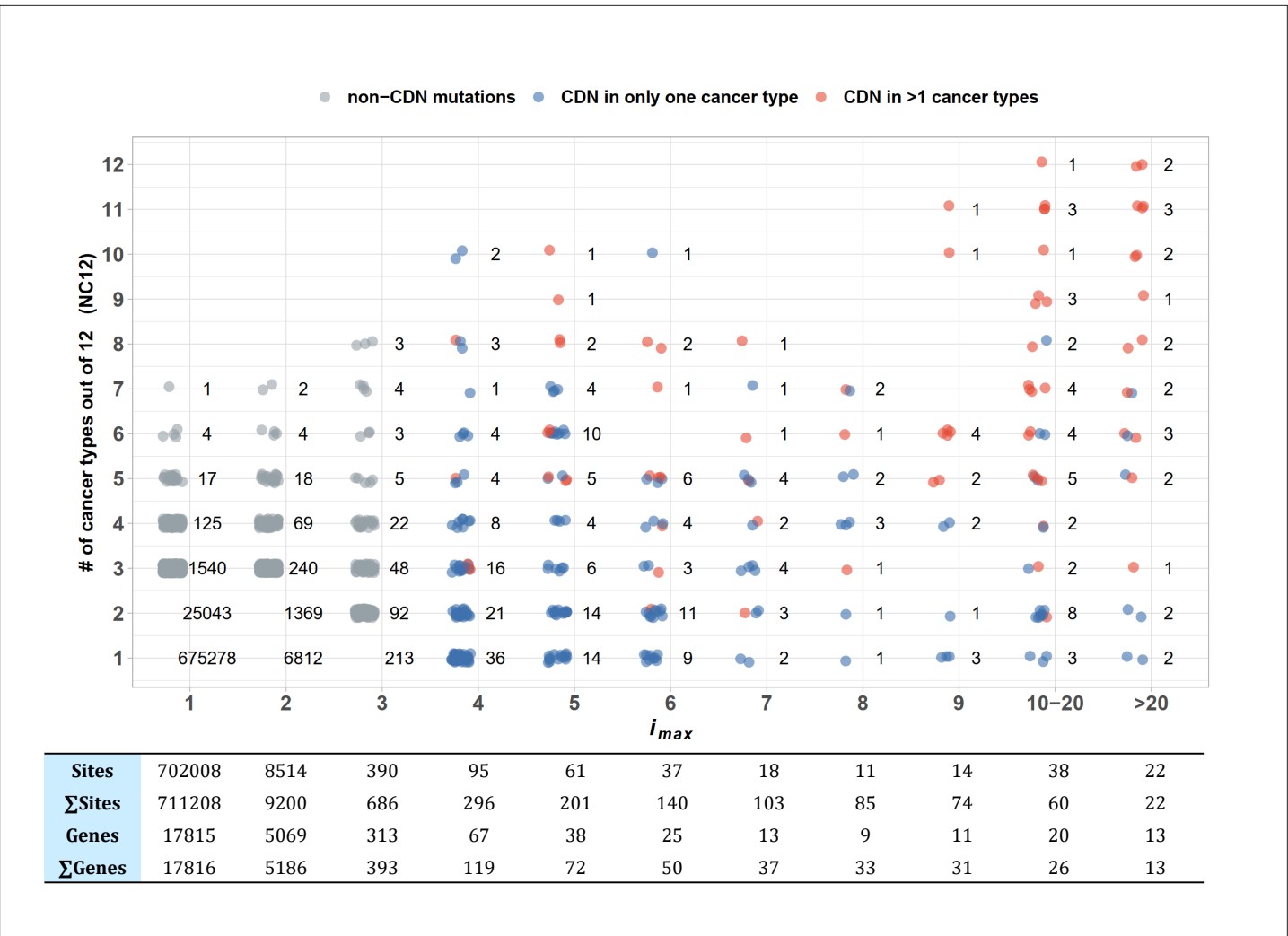

**Figure 4.** Sharing of cancer-driving nucleotides (CDNs) across cancer types. The X-axis shows $i_{max}$, which is the largest $i$ a CDN reaches among the 12 cancer types. The Y-axis shows the number of cancer types where the mutation also occurs. Each dot is a CDN, and the number of dots in the cloud is given. The blue and red dots denote, respectively, mutations classified as a CDN in one or multiple cancer types. Gray dots are non-CDNs. The table in the lower panel summarizes the number of sites and the number of genes harboring these sites.

types, with a median NC12 of 9. On the opposite end, two CDNs with $i_{max} > 20$ are observed in only one cancer type (*EGFR*: T2573 in lung and *FGFR2*: C755 in endometrium cancer). The bimodal pattern suggests that a few cancer driver mutations are tissue specific, whereas most others appear to have pan-cancer-driving potentials.

To conclude, when a driver is observed in more than one cancer type, it is often a cancer driver in many others, but insufficiently powerful to meet the statistical criteria for driver mutations. This pan-cancer property can be seen at the higher resolution of CDN, but is often missed at the whole-gene level. Cancers of the same tissue in different patients, often reported to have divergent mutation profiles (*Nik-Zainal et al., 2012*; *Roberts and Gordenin, 2014*), should be a good test of this hypothesis.

## CDNs in relation to individual patients and therapeutic strategies

In previous sections, the focus is on the population of cancer patients; for example, how many in the patient population have certain mutations. We now direct the attention to individual patients. It would be necessary to pinpoint the CDN mutations in each patient in order to delineate the specific

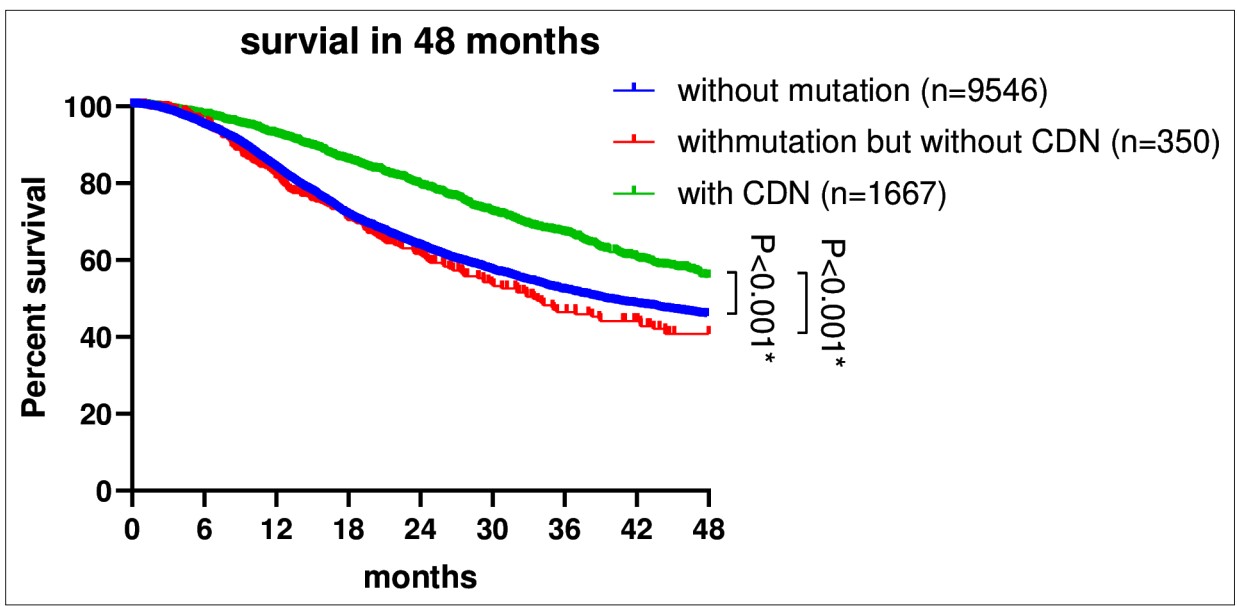

**Figure 5.** Survival analysis of non-small cell lung cancer (NSCLC) patients based on EGFR mutation status. Patient data were retrieved from the GENIE database (https://genie-public-beta.cbioportal.org/) and stratified into three groups based on *EGFR* mutation profiles: Group 1 comprises patients with *EGFR* CDN mutations; group 2 includes patients with nonsynonymous mutations in *EGFR* that are not cancer-driving nucleotides (CDNs); the *EGFR^{WT}* group consists of patients with no *EGFR* mutations (see 'Methods'). Patients of groups 1 and 2 received *EGFR*-targeted therapies in accordance with the guidelines for managing *EGFR* mutant NSCLC (*Passaro et al., 2022*; *Choudhury et al., 2023*). Survival analysis using the Kaplan–Meier method revealed a significantly higher survival rate for group 1 patients compared to group 2 and the *EGFR^{WT}* group (p<0.001).

evolutionary path and to devise the treatment strategy. We shall first address the cancer-driving power of CDN vs. non-CDN mutations in the same gene.

## Efficacy of targeted therapy against CDNs vs. non-CDNs

In general, a patient would have many point mutations, only a few of which are strong CDNs. We may ask whether most mutations on the canonical genes, such as *EGFR*, are CDNs. Presumably, synonymous, and likely many nonsynonymous, mutations on canonical genes may not be CDNs. It would be logical to hypothesize that patients whose *EGFR* has a CDN mutation (group 1 patients) should benefit from the gene-targeted therapy more than patients with a non-CDN mutation on the same gene (group 2 patients). In the second group, *EFGR* may be a nondriver of tumorigenesis.

Published data (*André et al., 2017*; *Choudhury et al., 2023*) are re-analyzed as shown in *Figure 5*. The hypothesis that patients of group 2 would not benefit as much as those of group 1 is supported by the analysis. This pattern further strengthens the underlying assumption that non-CDN mutations, even on canonical genes, are not cancer drivers.

## Number of CDNs in each patient

We postulate that a full set of CDNs should be able to inform about the cause of each cancer as well as the design of gene-targeted therapy. In *Table 4*, the known CDNs based on TCGA are tallied. Note that only a few CDNs fall on the canonical driver genes, whereas most CDNs fall on the nonconventional ones.

In most cancer types, 10–30% of patients, shown in the $n_0$ row of *Table 4*, have no known CDNs (and >50% among breast cancer patients). Hence, the current practice is to rely on missense mutations, regardless of CDNs or non-CDNs, on the canonical genes. The CDN column vs. the gene column in *Table 4* addresses this issue. For example, the CDN column suggests that 33% of lung cancer patients (the $n_0$ row) would not respond well to gene-targeted therapy, whereas the gene column shows only 5.3%. The difference is due to a higher, and likely inflated, detection rate of candidate drivers in the gene column. We suggest that patients who have a non-CDN mutation on a driver gene would not respond to the targeted therapy against that gene, as demonstrated in *Figure 5*. In

**Table 4.** Numbers of patients with cancer-driving nucleotides (CDNs) vs. number of patients with any non-synonymous mutations in the same genes.

| | Lung | | Breast | | Central nervous system | | Upper aerodigestive tract | | Colon | | Endometrium | |
|---|---|---|---|---|---|---|---|---|---|---|---|---|
| | CDN*[†] (178) | Gene[†] ‡ (89) | CDN (50) | Gene (17) | CDN (83) | Gene (24) | CDN (77) | Gene (32) | CDN (148) | Gene (96) | CDN (142) | Gene (56) |
| $n_0$ | 342 (33%) § | 53 (5.3%) | 492 (51.1%) | 415 (43.1%) | 235 (26.9%) | 163 (18.7%) | 268 (39%) | 140 (20.3%) | 102 (17.9%) | 42 (7.4%) | 42 (9%) | 14 (3%) |
| $n_1$ | 411 (39.7%) | 70 (6.8%) | 379 (39.4%) | 395 (41%) | 359 (41.1%) | 306 (35.1%) | 268 (39%) | 229 (33.3%) | 159 (27.8%) | 79 (13.8%) | 108 (23.2%) | 59 (12.7%) |
| $n_2$ | 192 (18.6%) | 84 (8.1%) | 73 (7.6%) | 114 (11.8%) | 225 (25.8%) | 293 (33.6%) | 101 (14.7%) | 171 (24.9%) | 140 (24.5%) | 93 (16.3%) | 169 (36.3%) | 101 (21.7%) |
| $n_{>2}$ | 90 (8.7%) | 826 (79.8%) | 18 (1.9%) | 38 (3.9%) | 53 (6.1%) | 110 (12.6%) | 50 (7.3%) | 147 (21.4%) | 170 (29.8%) | 357 (62.5%) | 146 (31.4%) | 291 (62.6%) |
| Total n | 1035 | 1035 | 963 | 963 | 873 | 873 | 688 | 688 | 571 | 571 | 465 | 465 |
| Mean # | 1.06 | 7.19 | 0.61 | 0.78 | 1.12 | 1.44 | 0.93 | 1.63 | 1.96 | 4.6 | 2.17 | 3.7 |

*$n_i$ designates the number of patients with $i$ CDN mutations.

[†]The number in the parentheses is the total number of CDNs or genes.

‡In this column, $n_i$ designates the number of patients with any nonsynonymous mutation in the same gene as the CDN column.

§There are 684 CDNs summed over all cancer types. The percentage is $n_i$/Total **n**.

the above example, 27.7% (33–5.3%) of patients may be subjected to the targeted treatment but may not respond well.

## Prevalence vs. potency of CDN-bearing genes in driving tumorigenesis

The last question is the relationship between mutation prevalence and tumorigenic strength (or potency) among CDN-bearing genes. For example, when a patient is diagnosed to have five CDNs in five genes, what may be their relative contributions to the tumorigenesis? Are they equally valid candidates for targeted therapy? It would seem logical that canonical CDGs with many CDNs should be the targets. However, because these genes would contribute at most one CDN to the tumorigenesis

**Table 5.** Gene numbers for different cancer hallmarks.

| | Gene number | | |
|---|---|---|---|
| Hallmark | All records | Breast | Colon |
| Angiogenesis | 78 | 8 | 6 |
| Cell division control | 107 | 12 | 10 |
| Cell replicative immortality | 44 | 4 | 3 |
| Change of cellular energetics | 70 | 10 | 4 |
| Escaping immune response to cancer | 51 | 1 | 1 |
| Escaping programmed cell death | 202 | 32 | 20 |
| Genome instability and mutations | 106 | 10 | 7 |
| Invasion and metastasis | 206 | 52 | 27 |
| Proliferative signaling | 176 | 40 | 20 |
| Senescence | 48 | 3 | 5 |
| Suppression of growth | 130 | 11 | 12 |
| Tumor-promoting inflammation | 54 | 2 | 3 |

Data downloaded from COSMIC (https://cancer.sanger.ac.uk/cosmic/download), see 'Methods'.

(*Figure 3B*), targeting a high-prevalence gene may not yield more benefits to the patients than targeting a low-prevalence gene that has a CDN.

The implication is that prevalence and potency of CDNs may not be strongly correlated. Some genes may be prevalently mutated in the patient population but, in each affected patient, these genes may not be more potent than the less prevalent genes with a CDN mutation. Potency can be tested in vitro by gene editing or in vivo by targeting treatment. In this interpretation, targeting a CDN of low prevalence (say, $i = 3$) may be as effective in treatment as targeting a high-prevalence CDN with $i = 20$. The model and *Table 5* present this hypothesis based on cancer hallmarks.

The hallmarks of cancer were first proposed by *Hanahan and Weinberg, 2000* with several updates (*Hanahan and Weinberg, 2011*; *Hanahan, 2022*). Each hallmark is a cancer phenotype shown in *Table 5* that lists the number of genes involved in each particular hallmark (see 'Methods'). While each hallmark may be associated with a number of genes, many genes are also involved in multiple hallmarks. As even the highly prevalent genes would usually have at most one mutation in each patient, we assume that each gene is associated with one hallmark in each patient.

Suppose that tumorigenesis requires a mutation in most (but perhaps not all) of the hallmarks, then the number of mutation combinations would be the product of all numbers in the corresponding column. For breast cancer, it would be $8 \times 12 \times 4.... \times 11 \times 2 – 1.7 \times 10^{11}$. In other words, the possible mutation combinations that can drive breast cancer is over a billion. Hence, two breast cancers are unlikely to have the same set of CDGs or CDNs. In this view, the prevalence of a gene would be inversely proportional to the hallmark gene number. For example, genes of '*invasion and metastasis*' in breast cancer would have a prevalence of <1/52. In contrast, the potency in tumorigenesis should depend on the hallmark phenotype itself and independent of gene number for that hallmark. In this example, each gene of '*invasion and metastasis*' may be lowly prevalent, but could also be highly potent in each patient.

In short, the prevalence and potency of CDNs may be poorly correlated. The hypothesis can be functionally tested (by gene editing in vitro or targeting treatment in vivo) in conjunction with the data on the attraction (i.e., co-occurrences) vs. repulsion (lack of co-occurrences) of CDNs.

## Discussion

The companion study presents the theory that computes the limit of recurrences ($i/\mathbf{n}$, $i$ times in $\mathbf{n}$ patients) of reachable by neutral mutations. Above the cutoff (e.g., 3/1000), a recurrent mutation is deemed an advantageous CDN (*Zhang et al., 2024*). At present, the power of CDN analysis is hampered by the still small sample sizes, generally between 300 and 3000. We show that, when $\mathbf{n}$ reaches $10^5$, a mutation only has to recur 12 times to be shown as a CDN, that is, 25 times more sensitive than 3/1000. In short, nearly all CDNs should be discovered with $\mathbf{n} \geq 10^5$.

In this study, we apply the theory on existing data to characterize the discovered CDNs. Based on TCGA data, this study concludes that each cancer patient carries only 1–2 CDNs, whereas 6–10 drivers are usually hypothesized to be present in each cancer genome (*Hanahan and Weinberg, 2011*; *Vogelstein et al., 2013*; *Campbell et al., 2020*). This deficit signifies the current incomplete understanding of cancer-driving potentials. Across patients of the same cancer type, about 50–150 CDNs have been discovered for each cancer type, representing perhaps only 10% of all possible CDNs. Given a complete set of CDNs, it should be possible to delineate the path of tumor evolution for each individual patient.

Direct functional test of CDNs would be to introduce putative cancer-driving mutations and observe the evolution of tumors. Such a task of introducing multiple mutations that are collectively needed to drive tumorigenesis has been done only recently and only for the best-known cancer-driving mutations (*Ortmann et al., 2015*; *Takeda et al., 2015*; *Hodis et al., 2022*). In most tumors, the correct combination of mutations needed is not known. Clearly, CDNs, with their strong tumorigenic strength, are suitable candidates.

Many CDNs in a patient may not fall on conventional CDGs, whereas these conventional CDGs may have passenger or weak mutations. Therefore, the efforts in gene-targeting therapy may well be shifted to the CDN-harboring genes. Given a complete set of CDNs, many more driver genes can be identified. Since many driver genes cannot be targeted for biological or technical reasons (*Dang et al., 2017*; *Danesi et al., 2021*; *Waarts et al., 2022*), a large set of CDGs will be desirable. The goal is that each cancer patient would have multiple targetable CDGs, all driven by CDNs they carry. In that

case, the probability that resistance mutations eluding multiple targeting drugs should be diminished (*Chen et al., 2022a*; *Zhai et al., 2022*; *Bian et al., 2023*; *Lin et al., 2023*; *Zhu et al., 2023*).

In this context, we should comment on the feasibility of targeting CDNs that may occur in either oncogenes (ONCs) or tumor suppressor genes (TSGs). It is generally accepted that ONCs drive tumorigenesis thanks to the gain-of-function (GOF) mutations, whereas TSGs derive their tumorigenic powers by loss-of-function (LOF) mutations. Nevertheless, since LOF mutations are likely to be widespread on TSGs, they are less likely to recur as CDNs. The even distributions of nonsense mutations along the length of many TSGs provide such evidence. Importantly, as gene targeting aims to diminish gene functions, GOF mutations are perceived to be targetable, whereas LOF mutations are not. By extension, ONCs should be targetable but TSGs are not, an assertion we address below.

The data suggest that missense mutations on TSGs may often be of the GOF kind. If missense mutations are far more prevalent than nonsense mutations in tumors, the missense mutations cannot possibly be LOF mutations. (After all, it is not possible to lose *more* functions than nonsense mutations.) In a separate study (*Deng et al., 2022*), we compare missense and nonsense mutations (referred to as the escape-route analysis). For example, AAA to AAC (K to Q) is a missense mutation while the same AAA codon to AAT (K to stop) is a nonsense mutation. We found many cases where the missense mutations on TSGs are more prevalent (>10×) than nonsense mutations. We interpret these missense mutations to be of the GOF kind because they could not possibly 'lose' more functions than the nonsense mutations.

Another interesting pattern may be the distributions of CDNs across different cancer types. Cancer evolution in different tissues represents parallel evolution driven by similar selection for cell proliferation but under different ecological conditions. *Figure 4* suggests that CDNs previously identified to be cancer-specific may have pan-cancer effects. In different cancer types, the same CDNs may drive tumorigenesis but the strength may not be sufficient to raise the data above the statistical threshold.

The CDN approach has two additional applications. First, it can be used to find CDNs in noncoding regions. Although the number of whole-genome sequences at present is still insufficient for systematic CDN detection, the preliminary analysis suggests that the density of CDNs in noncoding regions is orders of magnitude lower than in coding regions. Second, CDNs can also be used in cancer screening with the advantage of efficiency as the targeted mutations are fewer. For the same reason, the false-negative rate should be much lower too. Indeed, the false-positive rate should be far lower than the gene-based screen which often shows a false-positive rate of >50% (Appendix 2, 'The specificity of CDNs in cancer detection').

Cancer evolution falls within the realm of ultra-microevolution (*Wu et al., 2016*). The repeated evolution addresses the single most severe criticism of evolutionary studies, namely all evolutionary events have a sample size of one. Such repeated evolution offers the opportunity to uncover the full list of mutations underlying complex traits that is at the heart of molecular evolution. The genetics of speciation (*Wu and Ting, 2004*; *Pan et al., 2022*; *Wang et al., 2022*; *Wu, 2023*) and the emergence of major viral strains (such as COVID-19) (*Deng et al., 2022*; *Ruan et al., 2022*; *Cao et al., 2023*; *Ruan et al., 2023*) are both phenomena of complex gene interactions. The two companion studies may thus unite evolutionary biology and cancer medicine.

## Methods
### Data preparation

Single-nucleotide variant (SNV) data for TCGA patients were downloaded from the GDC Data Portal (https://portal.gdc.cancer.gov/, data version 28 February 2022), with mutations identified by at least two pipelines were included in this study. Mutations exceeding a 1‰ frequency in the Genome Aggregation Database (*gnomAD*, version v2.1.1) were excluded to minimize potential false positives arising from germline variants. Patients with more than 3000 coding region point mutations were filtered out as potential hypermutator phenotypes. This filtering process yielded a final analysis set encompassing 7369 patients across 12 diverse cancer types for subsequent analysis. The calculation of $A_i$ and $S_i$ follows the same method as described in the companion paper (*Zhang et al., 2024*).

For CDN analysis in noncancerous tissues, mutation profiles for normal tissues were retrieved from *SomaMutDB* (*Sun et al., 2022*). Mutations from different samples originating from the same individual were consolidated. Donners above the age of 80 were excluded from our dataset. The mutation

processing followed the same pipeline as previously described. In total, we have mutation profiles from 487 donners serving as a negative control.

The canonical lists of CDGs were obtained from three distinct data sources. The CGC Tier 1 genes, encompassing genes with the highest confidence of driver status, were retrieved from the COSMIC Cancer Gene Census (https://cancer.sanger.ac.uk/census; *Sondka et al., 2018*). The IntOGen driver gene list, which employs an integrated pipeline for gene discovery, was downloaded from https://www.intogen.org/download (*Martínez-Jiménez et al., 2020*). Bailey's driver gene list comprises 299 CDGs identified through a *PanSoftware* strategy, with further experimental validation confirming their role in driving cell lines (*Bailey et al., 2018*). The consistency of cancer types across all studies was manually verified using *oncotree* (#/home). For the analysis of driver gene overlap, only drivers from the same cancer type were compared.

The hallmark annotation of genes was downloaded from COSMIC (https://cancer.sanger.ac.uk/cosmic/download), encompassing 331 genes with annotated dysregulated biological processes. It is important to note that these hallmarks are manually annotated as part of an ongoing effort to characterize the role of genes in cancer based on literature evidence. The actual scale of hallmark genes may be substantially larger than the current version.

For gene-level selection analysis, we utilized the R package '*dndscv*' to quantify selection signals for missense and nonsense mutations in a given gene (*Martincorena et al., 2017*). Specifically, the package calculates the Ka/Ks ratio, denoted as '$w$' in the final results, for a given mutation impact (missense or nonsense). The significance of selection is presented as $q$ values after Benjamini–Hochberg (BH) adjustment. Genes with $w > 1$ and $q < 0.1$ were identified as being significantly under positive selection.

We employ $i^* = 3$ as a cutoff for identifying CDNs across various cancer types. The specific value of $i^*$ is detailed in Eq. 10 of the companion paper (*Zhang et al., 2024*). Here, $i^* = 3$ is chosen consistently across all cancer types, taking into account the abundance of sites under positive selection given $i = 3$ in *Table 2*. Throughout our analysis, emphasis is placed on CDNs of the missense category, where missense mutations with a recurrence $\geq 3$ are identified as CDNs. For $\Delta Ui$ analysis, the reference table for 75 single-step amino acid changes was obtained from *Chen et al., 2019a*, and the $\Delta Ui$ for each CDN is derived by mapping the amino acid change to the reference table.

## Calculation of $A_{i\_e}$

We employ Eq. 9 from the companion paper to calculate the expected value for $A_i$ under neutrality. For a given site, the cumulative probability for recurrence $x \leq i - 1$ could be expressed as

$$F\left(x \leq i - 1\right) = 1 - \left(1 - \frac{1}{1 + nE\left(u\right)}\right)^{i-1} \tag{S1}$$

where **n** is the population size of a given cancer type, and **E(u)** is the mutation rate per site per patient derived from singleton synonymous mutations:

$$S_1 = L_S \cdot nE\left(u\right) e^{-(n-1)E(u)} \tag{S2}$$

Then by expectation, site number of recurrence $i$ ($A_{x \geq i}$) could be represented by

$$A_{x \geq i} = L_A - L_A \cdot F\left(x \leq i - 1\right)$$

Following the same logic, we will have $A_{x \geq i+1}$ as

$$A_{x \geq i+1} = L_A - L_A \cdot F\left(x \leq i\right)$$

Then the expected value for $A_{i\_e}$ is

$$A_{i\_e} = A_{x \geq i} - A_{x \geq i+1} = L_A \cdot \left[F\left(x \leq i\right) - F\left(x \leq i - 1\right)\right] \tag{S3}$$

**$L_A$** and **$L_S$** are missense and synonymous sites, respectively. The calculation procedure is described in methods of the companion paper (*Zhang et al., 2024*).

With *Equation S1*, *Equation S2*, *Equation S3*, we could solve for the expected number of sites with missense mutation recurrence *i*.

## Survival analysis of *EGFR*-targeted therapy

The mutation and clinical profiles of 23,253 patients were retrieved from the GENIE project (*Cerami et al., 2012*; *de Bruijn et al., 2023*), with 7216 patients harboring *EGFR* mutations. Survivor months were calculated as the time elapsed between the date of sequencing and the date of the last contact (or day of death). In cases where patients had multiple sequencing reports, the earliest one was selected. For CDN calling, we applied Eq. 10 from the companion paper (*Zhang et al., 2024*). With $\varepsilon = 0.01$, we set the CDN cutoff $i^* = 14$. To mitigate potential biases from other common drivers in lung cancer, patients with indels in exons 19 and 20 of *EGFR*, G12/13 mutations in *KRAS*, V600 mutations in *BRAF*, exon 20 insertions in *HER2*, fusions in *MET*, *ALK*, *ROS1*, *RET*, *NTRK*, and *MET* were filtered out. The final survival analysis was conducted using GraphPad Prism 8.

## Annotation for noncanonical CDN genes

We conducted functional annotation and enrichment analysis for newly identified noncanonical CDN genes using four independent databases (Gene Ontology, KEGG, Disease Ontology, and Reactome) with R packages (*clusterProfiler*, *DOSE*, *ReactomePA*). For each analysis, we set a p-value cutoff of 0.05 and a q-value cutoff of 0.2, with p-value adjustment method set to 'BH'. To explore the connections between noncanonical CDN genes and canonical CDGs, enrichment analyses were performed alongside cancer drivers from IntOGen. Specifically, for enrichment annotations related to cancer hallmarks, the corresponding genes were subjected to manual confirmation using CancerGeneNET (https://signor.uniroma2.it/CancerGeneNet/).

## Acknowledgements

We wish to acknowledge the supports from the First Affiliated Hospital, the Seventh Affiliated Hospital of Sun Yat-sen University, Cancer Center of Clifford Hospital, Jinan University, Cancer Hospital Chinese Academy of Medical Sciences, Shenzhen Center, and Guangdong Academy of Medical Sciences, Guangdong Provincial People's Hospital on the startup of the Cancer Driving Nucleotide (CDN) project. We would like to acknowledge Kunming Institute of Zoology for discussing the ideas of CDN. We thank Weiwei Zhai, Qianfei Wang, and Weini Huang for insightful comments and suggestions. We would also like to acknowledge the American Association for Cancer Research (AACR) and The Cancer Genome Atlas (TCGA) project, which have provided invaluable datasets and resources that have significantly enriched our understanding of cancer biology and improved patient outcomes. This work was supported by the National Natural Science Foundation of China (32150006, 32293193, 32293190, 32370659, and 32200493) to CIW and 82341092 to HJW, the National Key Research and Development Projects of the Ministry of Science and Technology of China (2021YFC0863300, 2021YFC0863400), Guangdong Key Research and Development Program (no. 2022B1111030001), and Guangdong Basic and Applied Basic Research Foundation (no. 2023A1515010016).

## Additional information

### Funding

| Funder | Grant reference number | Author |
|---|---|---|
| National Natural Science Foundation of China | 32150006 | Chung-I Wu |
| Guangdong Key R&D Project of China | 2022B1111030001 | Hai-Jun Wen |
| National Natural Science Foundation of China | 32293193 | Chung-I Wu |
| National Natural Science Foundation of China | 32293190 | Chung-I Wu |

| Funder | Grant reference number | Author |
|--------|------------------------|--------|
| National Natural Science Foundation of China | 82341092 | Hai-Jun Wen |
| National Key Research and Development Program of China | 2021YFC0863300 | Chung-I Wu |
| National Key Research and Development Program of China | 2021YFC0863400 | Chung-I Wu |
| National Natural Science Foundation of China | 32200493 | Chung-I Wu |
| National Natural Science Foundation of China | 32370659 | Chung-I Wu |
| Guangdong Basic and Applied Basic Research Foundation | 2023A1515010016 | Chung-I Wu |

The funders had no role in study design, data collection and interpretation, or the decision to submit the work for publication.

## Author contributions

Lingjie Zhang, Conceptualization, Data curation, Formal analysis, Validation, Investigation, Visualization, Methodology, Writing – original draft; Tong Deng, Changhao Shi, Validation, Visualization; Zhongqi Liufu, Xiangnyu Chen, Shijie Wu, Bingjie Chen, Data curation, Validation; Xueyu Liu, Data curation, Validation, Visualization; Zheng Hu, Resources, Investigation, Project administration; Qichun Cai, Resources, Validation, Investigation; Chenli Liu, Resources, Supervision, Investigation, Project administration; Mengfeng Li, Resources, Supervision, Project administration; Miles E Tracy, Writing – review and editing; Xuemei Lu, Conceptualization, Supervision, Investigation, Project administration; Chung-I Wu, Conceptualization, Resources, Supervision, Funding acquisition, Validation, Investigation, Project administration, Writing – review and editing; Hai-Jun Wen, Funding acquisition, Validation, Investigation, Methodology

## Author ORCIDs

Lingjie Zhang https://orcid.org/0000-0002-6506-4457
Xiangnyu Chen https://orcid.org/0000-0001-5078-8906
Zheng Hu https://orcid.org/0000-0003-1552-0060
Xuemei Lu https://orcid.org/0000-0001-6044-6002
Chung-I Wu https://orcid.org/0000-0001-7263-4238
Hai-Jun Wen https://orcid.org/0000-0001-8676-1254

Reviewer #1 (Public review): https://doi.org/10.7554/eLife.99341.3.sa1
Reviewer #2 (Public review): https://doi.org/10.7554/eLife.99341.3.sa2
Author response https://doi.org/10.7554/eLife.99341.3.sa3

# Additional files

## Supplementary files
• MDAR checklist

## Data availability

The scripts for generating the key results of this study and the accompanying paper (*Zhang et al., 2024*) are available at GitLab (copy archived at *Zhang, 2024*). Example files for breast cancer analysis have also been included. The complete set of CDNs can be found in Supplementary file 1 of the accompanying paper (*Zhang et al., 2024*).

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

## Appendix 1

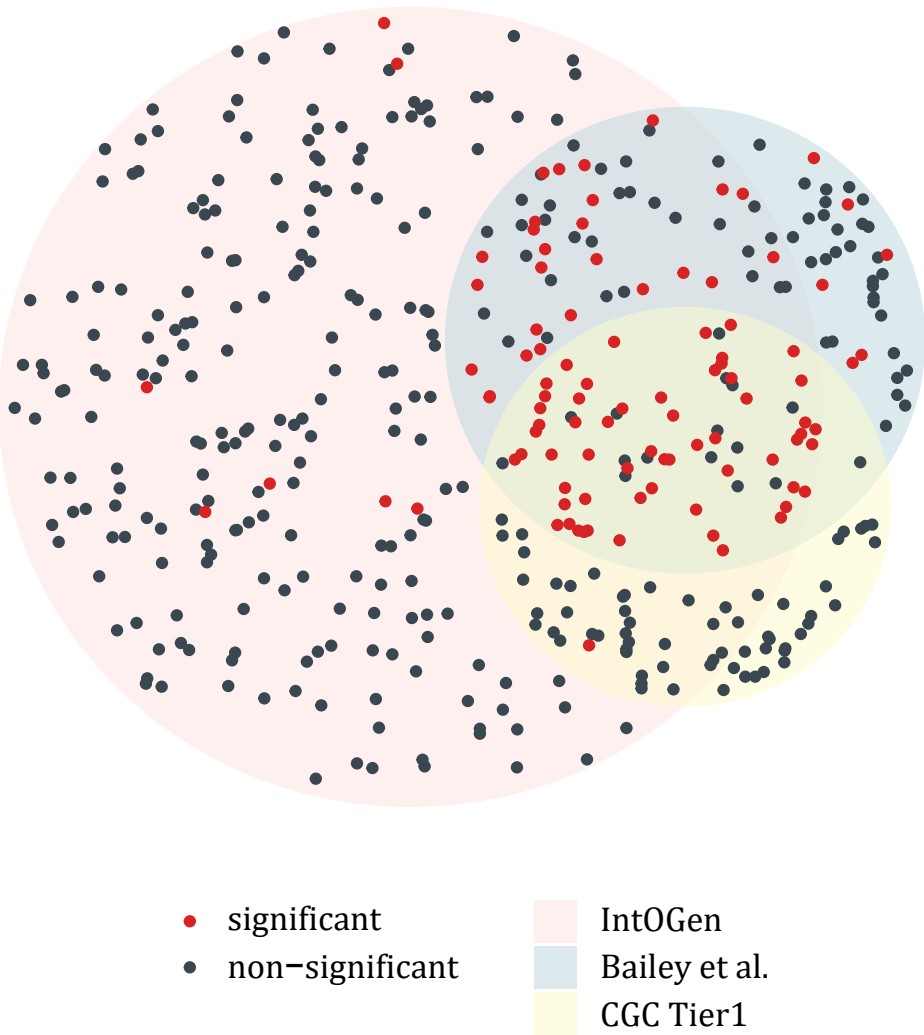

**Appendix 1—figure 1.** The overlap of cancer drivers from IntOGen, Bailey et al. and CGC Tier 1 (*Bailey et al., 2018*; *Sondka et al., 2018*; *Martínez-Jiménez et al., 2020*). Driver genes (dots) for 12 cancer types were extracted from each driver list, indicated by three different region colors. The area size of each region is proportional to the gene number, with 384 genes for IntOGen, 168 for Bailey et al. and 137 for CGC Tier 1. Genes with a significant positive selection signal in the merged mutation set are marked in red, while nonsignificant ones are colored in blue. Notably, genes shared across the three driver sets are largely those with a significant Ka/Ks > 1.

## Appendix 2

### 1. Quantifying evolutionary fitness of CDN

We leverage Eqs. 2 and 4 from the companion paper (*Zhang et al., 2024*) and rewrite $A_i$ as follows:

$$A_i = A_i^{neut} + A_i^*  \tag{S4}$$

where $A_i$ represents the observed site number with missense recurrence of $i$, which could be further decomposed into two components: $A_i^{neut}$, the site number with missense recurrence of $i$ under neutral mutational force, and $A_i^*$, which occurs under positive selection. For $A_i^*$, we have

$$A_i^* = f \cdot L_A \cdot g\left(i, k\right) \left[w \cdot nE\left(u\right)\right]^i  \tag{S5}$$

With *Equations S4 and S5*, $A_i$ could be expressed as

$$A_i \sim \frac{\Gamma\left(k+i\right)}{\Gamma\left(i+1\right)\Gamma\left(k\right)} \cdot \frac{L_A}{k^i}\left[nE\left(u\right)\right]^i \cdot \left(1 + f \cdot w^i\right) = 2.3 \cdot S_i \cdot \left(1 + f \cdot w^i\right)  \tag{S6}$$

where $f$ denotes the fraction of missense sites under positive selection ($f \ll 1$), and $w$ represents the selective advantage ($s$) scaled by the population size of progenitor cancer cell ($N$). First, we aimed to estimate $f$ from the discrepancy between $R_0 = A_0/S_0$ and $R_1 = A_1/S_1$. The number of sites under positive selection in $A_1$ ($A_1^*$) could be approximated from $A_0$ as $A_0 \cdot f$, which could also be derived from the excess of mutations from $A_1$ as $A_1 \cdot \left(R_1 - R_0\right)/R_1$, then we have

$$A_0 \cdot f = A_1 \cdot \left(R_1 - R_0\right)/R_1$$

$$f = \frac{A_1}{A_0} \cdot \left(1 - \frac{R_0}{R_1}\right)  \tag{S7}$$

Based on the average statistics in *Table 1*, $f$ could be estimated from *Equation S6* to be $3.13 \times 10^{-4}$. With synonymous recurrence sites as neutral reference, $A_i \sim 2.3 \cdot S_i \cdot \left(1 + f \cdot w^i\right)$. Given $A_3/S_3$ = 5.25, we will have $w$ = 16, which means $A_3$ would be 4096 ($16^3$) fold higher than the neutral expectation.

### 2. Functional annotation of new cancer drivers

A limitation in cancer driver discovery lies in the modeling of background mutation process, which often necessitates a balance between current knowledge and unknown mutational mechanisms. Consequently, genes recognized as canonical drivers in one cancer type may be categorized as noncanonical in others due to the lack of statistical significance. Of the 229 noncanonical drivers identified across six cancer types, 19 genes have been previously recognized as canonical drivers in different cancer types, while 23 genes were classified as drivers in IntOGen through a combination of diverse statistical methods.

For the newly identified noncanonical CDN genes in this study, we explore their potential functional relevance to cancer through a two-step procedure. First, we annotate these genes considering gene ontology, pathway, disease association, and protein–protein interaction with known cancer drivers. Subsequently, we conduct manual curation by reviewing published literatures for evidence related to cancer. *Appendix 2—figure 1* illustrates how noncanonical CDGs are enriched in cancer-related biological processes in lung and colon cancers. The results reveal that processes such as cell migration/adhesion, epithelial-to-mesenchymal transition (EMT), cell proliferation, energy metabolism, immune response, and DNA transcription activity are among the most enriched processes in both cancer types. Additionally, other cancer hallmark-related processes, such as the cell cycle control, DNA stability, and response to stresses, are also enriched among noncanonical CDN genes.

In *Appendix 2—figure 2*, we present the top 10 genes being most connected to known cancer drivers across four independent enrichment analyses. Here, we take two unidentified driver genes for example to illustrate their functional roles relating to cancer. *PIK3R2* (phosphoinositide-3-kinase regulatory subunit 2), which encodes p85β of class I PI3K, is often highly expressed in most tumors (*Liu et al., 2022*). This gene has a CDN mutation of G1117A (with amino acid change of G373R) in endometrium cancer, which is also presented in lung and urinary tract cancers. *PIK3R2*

has been reported as an oncogene, with its overexpression triggering cell transformation in culture and promoting cancer progression in mouse model (*Vallejo-Díaz et al., 2019*). *SLC7A5* (solute carrier family 7 member 5), with a CDN mutation of G1480A (V494I) in colon cancer, plays a critical oncogenic role in maintaining intracellular amino acid levels for an elevated protein synthesis rate in *KRAS*-mutant cells. Depletion of *SLC7A5* suppresses intestinal tumorigenesis in mice and resensitizes tumors to protein synthesis inhibition (*Najumudeen et al., 2021*). In conclusion, although excluded from the canonical driver list due to the lack of statistical significance, noncanonical CDN genes may still undergo positive selection at the site level. Ongoing research may provide further experimental evidence for these genes as part of an ongoing effort to identify the complete set of cancer drivers.

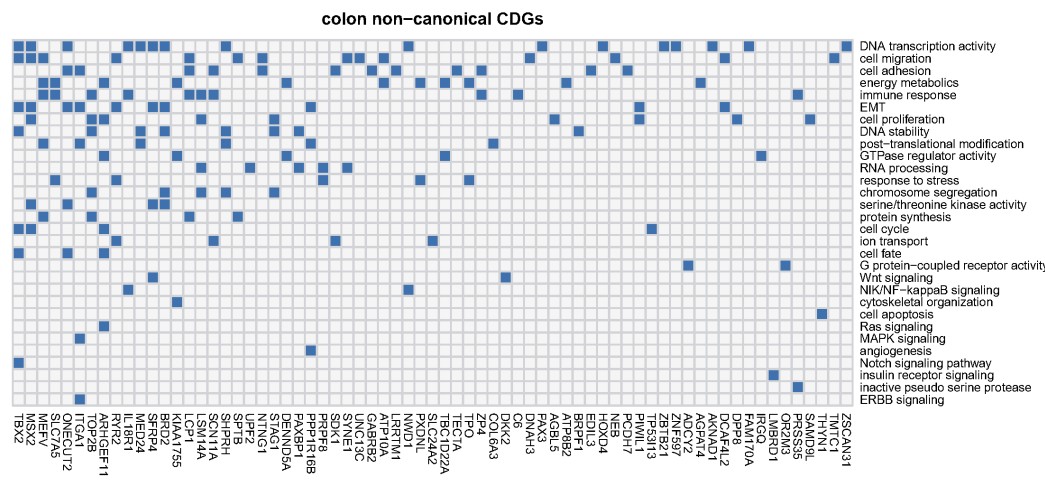

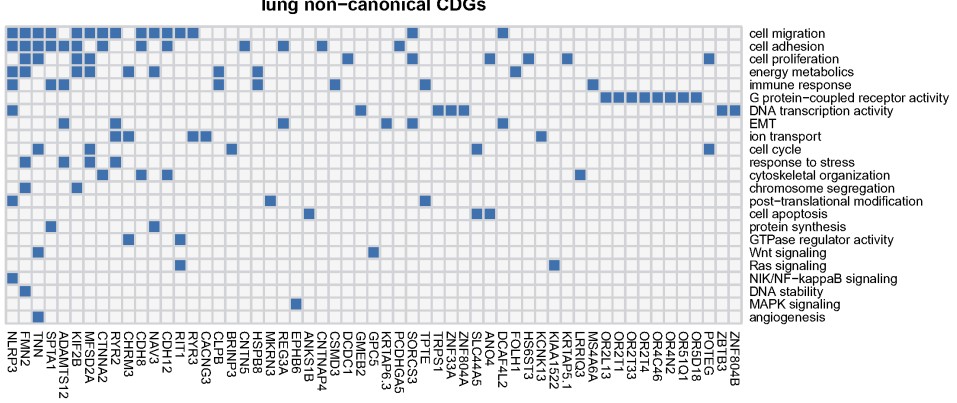

**Appendix 2—figure 1.** Noncanonical cancer driver genes (CDGs) in colon and lung cancer along with associated biological processes (Y-axis). For each gene, we examine its annotation results from GO analysis and search for cancer-related evidence in the literature. Biological processes are summarized and curated in relation to cancer hallmarks. Each connection between gene ID and biological process is depicted by a blue block in the grid.

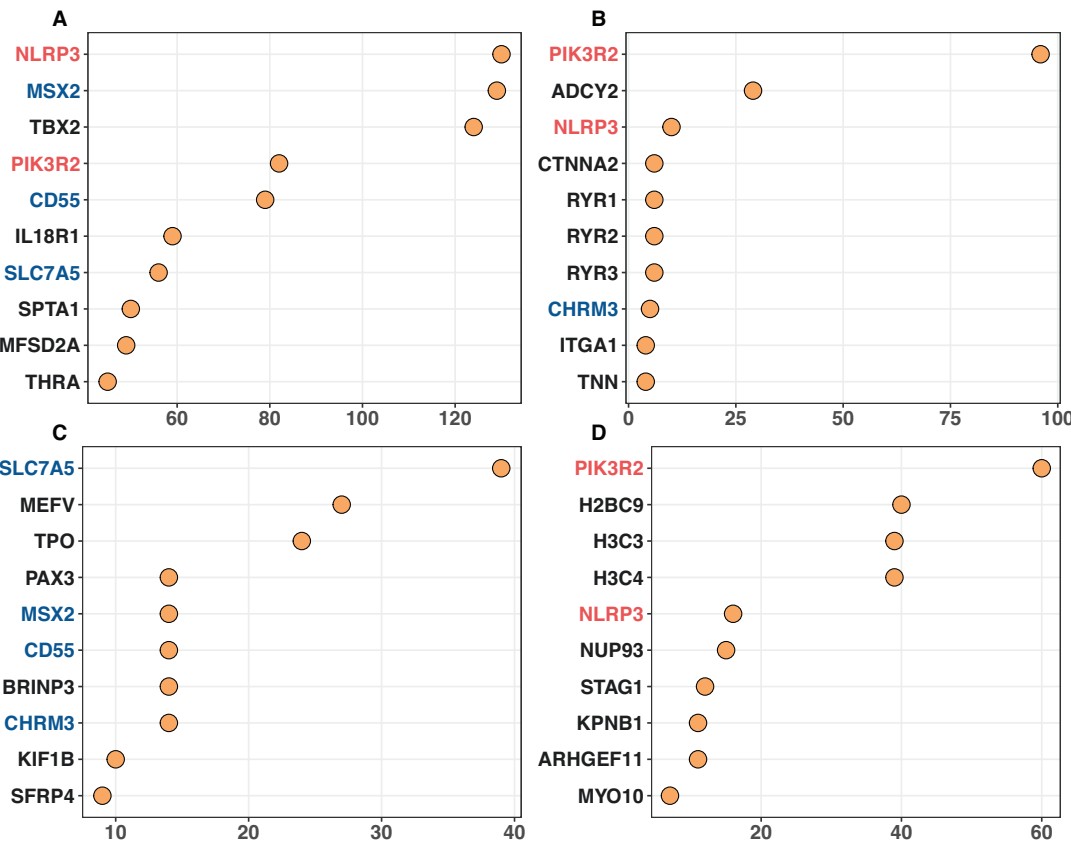

**Appendix 2—figure 2.** Top 10 noncanonical cancer driver genes (CDGs) with the highest enrichment records with IntOGen's driver list from four enrichment analysis. Panels (**A–D**) corresponds to Gene Ontology, KEGG, Disease Ontology, and Reactome analysis, respectively. The X-axis represents the number of enrichment records for each gene, while genes are listed on the Y-axis according to their enrichment record number. Genes with different occurrences across the top set of four analysis are marked with red (three hits), blue (two hits) and black (one hit).

## 3. The specificity of CDNs in cancer detection

Generally, in a given sample, each mutated gene would harbor one mutation. Therefore, we measure the false-positive rate for CDNs (or CDGs) as the proportion of individuals harboring nonsynonymous mutations at CDN (or CDGs). Across 487 individuals from noncancerous set, 446 are devoid of any mutations at CDNs, yielding a specificity of 91.6% (false-positive rate: 8.4%). In contrast, for CDGs downloaded from CGC, the specificity is 37.6% (false-positive rate: 62.4%). The high specificity implies the potential application of CDNs in biopsy and companion diagnostics, as well as the possibility of integration with other early screening pipelines. Furthermore, when we compared the mutations of recurrences $i \geq 3$ in *SomaMutDB* with CDNs identified in our analysis, no overlap was observed. The high exclusiveness of CDNs between cancer and noncancer implies that positive selection operates in a specific manner in cancer, distinct from normal tissues.

