## [Editor Report · eLife Assessment]

This **valuable** study is a companion to a paper introducing a theoretical framework and methodology for identifying cancer-driving nucleotides (CDNs). The evidence that recurrent SNVs or CDNs are common in true cancer driver genes is **convincing**, with more limited evidence that many more undiscovered cancer driver mutations will have CDNs, and that this approach could identify these undiscovered driver genes with about 100,000 samples.

---

## [Referee Report · Reviewer #1 (Public review)]

The study investigates Cancer Driving Nucleotides (CDNs) using the TCGA database, finding that these recurring point mutations could greatly enhance our understanding of cancer genomics and improve personalized treatment strategies. Despite identifying 50-150 CDNs per cancer type, the research reveals that a significant number remain undiscovered, limiting current therapeutic applications, underscoring the need for further larger-scale research.

Strengths:

The study provides a detailed examination of cancer-driving mutations at the nucleotide level, offering a more precise understanding than traditional gene-level analyses. The authors found a significant number of CDNs remain undiscovered, with only 0-2 identified per patient out of an expected 5-8, indicating that many important mutations are still missing. The study indicated that identifying more CDNs could potentially significantly impact the development of personalized cancer therapies, improving patient outcomes.

Weaknesses:

The challenges in direct functional testing of CDNs due to the complexity of tumor evolution and unknown mutation combinations limit the practical applicability of the findings.

---

## [Referee Report · Reviewer #2 (Public review)]

Summary:

The study proposes that many cancer driver mutations are not yet identified but could be identified if they harbor recurrent SNVs. The paper leverages the analysis from Paper #1 that used quantitative analysis to demonstrate that SNVs or CDNs seen 3 or more times are more likely due to selection (ie a driver mutation) than by chance or random mutation.

Strengths:

Empirically, mutation frequency is an excellent marker of a driver gene because canonical driver mutations typically have recurrent SNVs. Using the TCGA database, the paper illustrates that CDNs can identify canonical driver mutations (Fig 3) and that most CDN are likely to disrupt protein function (Fig 2). In addition, CDNs can be shared between cancer types (Fig 4).

Weaknesses:

Driver alteration validation is difficult, with disagreements on what defines a driver mutation, and how many driver mutations are present in a cancer. The value proposed by the authors is that the identification of all driver genes can facilitate the design of patient specific targeting therapies, but most targeted therapies are already directed towards known driver genes. There is an incomplete discussion of oncogenes (where activating mutations tend to target a single amino acid or repeat) and tumor suppressor genes (where inactivating mutations may be more spread across the gene). Other alterations (epigenetic, indels, translocations, CNVs) would be missed by this type of analysis.

The method could be more valuable when applied to the noncoding genome, where driver mutations in promoters or enhancers are relatively rare, or as yet to be discovered. Increasingly more cancers have had whole genome sequencing. Compared to WES, criteria for driver mutations in noncoding regions are less clear, and this method could potentially provide new noncoding driver CDNs. Observing the same mutation in more than one cancer specimen is empirically unusual, and the authors provide a solid quantitative analysis that indicates many recurrent mutations are likely to be cancer-driver mutations.

---

## [Author Response]

The following is the authors’ response to the original reviews.

**eLife Assessment**
This valuable study is a companion to a paper introducing a theoretical framework and methodology for identifying Cancer Driving Nucleotides (CDNs). While the evidence that recurrent SNVs or CDNs are common in true cancer driver genes is solid, the evidence that many more undiscovered cancer driver mutations will have CDNs, and that this approach could identify these undiscovered driver genes with about 100,000 samples, is limited.Same criticism as in the eLife assessment of eLife-RP-RA-2024-99340 (https://elifesciences.org/reviewed-preprints/99340). Hence, please refer to the responses to the companion paper.
**Public Reviews:**

**Reviewer #1 (Public Review):**
The study investigates Cancer Driving Nucleotides (CDNs) using the TCGA database, finding that these recurring point mutations could greatly enhance our understanding of cancer genomics and improve personalized treatment strategies. Despite identifying 50-150 CDNs per cancer type, the research reveals that a significant number remain undiscovered, limiting current therapeutic applications, and underscoring the need for further larger-scale research.Strengths:The study provides a detailed examination of cancer-driving mutations at the nucleotide level, offering a more precise understanding than traditional gene-level analyses. The authors found a significant number of CDNs remain undiscovered, with only 0-2 identified per patient out of an expected 5-8, indicating that many important mutations are still missing. The study indicated that identifying more CDNs could potentially significantly impact the development of personalized cancer therapies, improving patient outcomes.Weaknesses:The study is constrained by relatively small sample sizes for each cancer type, which reduces the statistical power and robustness of the findings. ICGC and other large-scale WGS datasets are publicly available but were not included in this study.

Thanks. We indeed have used all public data, including GENIE (figure 7 of the companion paper), ICGC and other integrated resources such as COSMIC. The main study is based on TCGA because it is unbiased for estimating the probability of CDN occurrences. In many datasets, the numerators are given but the denominators are not (the number of patients with the mutation / the total number of patients surveyed). In GENIE, we observed that E(u) estimated upon given sequencing panels are much smaller than in TCGA, this might be due to the selective report of nonsynonymous mutations for synonymous mutations are generally considered irrelevant in tumorigenesis.

To be able to identify rare driver mutations, more samples are needed to improve the statistical power, which is well-known in cancer research. The challenges in direct functional testing of CDNs due to the complexity of tumor evolution and unknown mutation combinations limit the practical applicability of the findings.

We fully agree. We now add a few sentences, making clear that the theory allows us to see how much more can be gained by each stepwise increase in sample size. For example, when the sample size reaches 106, further increases will yield almost no gain in confidence of CDNs identified see figures of eLife-RP-RA-2024-99340. As pointed out in our provisional responses, an important strength of this pair of studies is that the results are testable. The complexity is the combination of mutations required for tumorigenesis and the identification of such combinations is the main goal and strength of this pair of studies. We add a few sentences to this effect.

While the importance of large sample sizes in identifying cancer drivers is well-recognized, the analytical framework presented in the companion paper (https://elifesciences.org/reviewed-preprints/99340) goes a step further by quantitatively elucidating the relationship between sample size and the resolution of CDN detection.

The question is very general as it is about multigene interactions, or epistasis. The challenges are true in all aspects of evolutionary biology, for example, the genetics of reproductive isolation(Wu and Ting 2004). The issue of epistasis is difficult because most, if not all, of the underlying mutations have to be identified in order to carry out functional tests. While the full identification is rarely feasible, it is precisely the objective of the CDN project. When the sample size increases to 100,000 for a cancer type, all point mutations for that cancer type should be identifiable.

The QC of the TCGA data was not very strict, i.e, "patients with more than 3000 coding region point mutations were filtered out as potential hypermutator phenotypes", it would be better to remove patients beyond +/- 3*S.D from the mean number of mutations for each cancer type. Given some point mutations with >3 hits in the TCGA dataset, they were just false positive mutation callings, particularly in the large repeat regions in the human genome.

Thanks. The GDC data portal offers data calls from multiple pipelines, enabling us to select mutations detected by at least two pipelines. While including patients with hypermutator phenotypes could introduce potential noise, as shown in Eq. 10 of the main text, our method for defining the upper limit of i* is relative robust to the fluctuations in the E(u) of the corresponding cancer population. Since readers may often ask about this, we expand the Methods section somewhat to emphasize this point.

The codes for the statistical calculation (i.e., calculation of Ai_e, et al) are not publicly available, which makes the findings hard to be replicated.

We have now updated the section of “Data Availability” in both papers. The key scripts for generating the major results are available at: https://gitlab.com/ultramicroevo/cdn_v1.

**Reviewer #2 (Public Review):**
Summary:The study proposes that many cancer driver mutations are not yet identified but could be identified if they harbor recurrent SNVs. The paper leverages the analysis from Paper #1 that used quantitative analysis to demonstrate that SNVs or CDNs seen 3 or more times are more likely to occur due to selection (ie a driver mutation) than they are to occur by chance or random mutation.Strengths:Empirically, mutation frequency is an excellent marker of a driver gene because canonical driver mutations typically have recurrent SNVs. Using the TCGA database, the paper illustrates that CDNs can identify canonical driver mutations (Figure 3) and that most CDNs are likely to disrupt protein function (Figure 2). In addition, CDNs can be shared between cancer types (Figure 4).Weaknesses:Driver alteration validation is difficult, with disagreements on what defines a driver mutation, and how many driver mutations are present in a cancer. The value proposed by the authors is that the identification of all driver genes can facilitate the design of patient-specific targeting therapies, but most targeted therapies are already directed towards known driver genes. There is an incomplete discussion of oncogenes (where activating mutations tend to target a single amino acid or repeat) and tumor suppressor genes (where inactivating mutations may be more spread across the gene). Other alterations (epigenetic, indels, translocations, CNVs) would be missed by this type of analysis.

The above paragraph has three distinct points. We shall respond one by one.

First, … *can facilitate the design of patient-specific targeting therapies, but most targeted therapies are already directed towards known driver genes…*

We state in the text of Discussion the following that shows only a few best-known driving mutations have been targeted. It is accurate to say that < 5% of CDNs we have identified are on the current targeting list. Furthermore, this list we have compiled is < 10% of what we expect to find.

Direct functional test of CDNs would be to introduce putative cancer-driving mutations and observe the evolution of tumors. Such a task of introducing multiple mutations that are collectively needed to drive tumorigenesis has been done only recently, and only for the best-known cancer driving mutations (Ortmann et al. 2015; Takeda et al. 2015; Hodis et al. 2022). In most tumors, the correct combination of mutations needed is not known. Clearly, CDNs, with their strong tumorigenic strength, are suitable candidates.

Second, “There is an incomplete discussion of oncogenes (where activating mutations tend to target a single amino acid or repeat) and tumor suppressor genes (where inactivating mutations may be more spread across the gene).”

We sincerely thank the reviewer for this insightful comment. Below are two new paragraphs in the Discussion pertaining to the point:

In this context, we should comment on the feasibility of targeting CDNs that may occur in either oncogenes (ONCs) or tumor suppressor genes (TSGs). It is generally accepted that ONCs drive tumorigenesis thanks to the gain-of-function (GOF) mutations whereas TSGs derive their tumorigenic powers by loss-of-function (LOF) mutations. It is worthwhile to point out that, since LOF mutations are likely to be more widespread on a gene, CDNs are biased toward GOF mutations. The often even distribution of non-sense mutations along the length of TSGs provide such evidence. As gene targeting aims to diminish gene functions, GOF mutations are perceived to be targetable whereas LOF mutations are not. By extension, ONCs should be targetable but TSGs are not. This last assertion is not true because mutations on TSGs may often be of the GOF kind as well.

The data often suggest that mis-sense mutations on TSGs are of the GOF kind. If mis-sense mutations are far more prevalent than nonsense mutations in tumors, the mis-sense mutations cannot possibly be LOF mutations. (After all, it is not possible to lose *more* functions than nonsense mutations.) For example, AAA to AAC (K to Q) is a mis-sense mutation while AAA to AAT (K to stop) is a non-sense mutation. In a separate study (referred to as the escape-route analysis), we found many cases where the mis-sense mutations on TSGs are more prevalent (> 10X) than nonsense mutations. Another well-known example is the distribution of non-sense mutations TSGs. For example, on *APC*, a prominent TSG, non-sense mutations are far more common in the middle 20% of the gene than the rest (Zhang and Shay 2017; Erazo-Oliveras et al. 2023). The pattern suggests that even these non-sense mutations could have GOF properties.

The following response is about the clinical implications of our CDN analysis. Canonical targeted therapy often relies on the Tyrosine Kinase Inhibitors (TKIs) (Dang et al. 2017; Danesi et al. 2021; Waarts et al. 2022). Theoretically, any intervention that suppresses the expression of gain-of-function (GOF) CDNs could potentially have therapeutic value in cancer treatment. This leads us to a discussion of oncogenes versus TSGs in the context of GOF / LOF (loss of function) mutations. Not all mutations on oncogenes have oncogenic effect, besides, truncated mutations in oncogenes are often subject to negative selection (Bányai et al. 2021), the identification of CDNs within oncogenes is therefore crucial for developing effective cancer treatment guidelines. Secondly, while TSGs are generally believed to promote cancer development via loss of function mutations, research suggests that certain mutations within TSGs can have GOF-like effect, such as the dominant negative effect of truncated TP53 mutations (Marutani et al. 1999; de Vries et al. 2002; Gerasimavicius et al. 2022). Characterizing driver mutations as GOF or LOF mutations could potentially expand the scope of targeted cancer therapy. We’ll address this issue in a third study in preparation.

The method could be more valuable when applied to the noncoding genome, where driver mutations in promoters or enhancers are relatively rare, or as yet to be discovered. Increasingly more cancers have had whole genome sequencing. Compared to WES, criteria for driver mutations in noncoding regions are less clear, and this method could potentially provide new noncoding driver CDNs. Observing the same mutation in more than one cancer specimen is empirically unusual, and the authors provide a solid quantitative analysis that indicates many recurrent mutations are likely to be cancer-driver mutations.

Again, we are grateful for the comments which prompt us to expand a paragraph in Discussion, reproduced below.

The CDN approach has two additional applications. First, it can be used to find CDNs in non-coding regions. Although the number of whole genome sequences at present is still insufficient for systematic CDN detection, the preliminary analysis suggests that the density of CDNs in non-coding regions is orders of magnitude lower than in coding regions. Second, CDNs can also be used in cancer screening with the advantage of efficiency as the targeted mutations are fewer. For the same reason, the false negative rate should be much lower too. Indeed, the false positive rate should be far lower than the gene-based screen which often shows a false positive rate of >50% (supplement File S1).

Again, we are grateful that Reviewer #2 have addressed the potential value of our study in finding cancer drivers in non-coding regions. A major challenge in this area lies in defining the appropriate *L* value as presented in Eq. 10. In the main text, we used a gamma distribution to account for the variability of mutation rates across sites in coding region. For the non-coding region, we will categorize these regions based on biological annotations. The goal is to set different *i** cutoffs for different genomic regions (such as heterochromatin / euchromatin, GC-rich regions or centromeric regions), and avoid false positive calls for CDN in repeated regions (Elliott and Larsson 2021; Peña et al. 2023).

References

Bányai L, Trexler M, Kerekes K, Csuka O, Patthy L. 2021. Use of signals of positive and negative selection to distinguish cancer genes and passenger genes. *Elife* 10:e59629.

Danesi R, Fogli S, Indraccolo S, Del Re M, Dei Tos AP, Leoncini L, Antonuzzo L, Bonanno L, Guarneri V, Pierini A, et al. 2021. Druggable targets meet oncogenic drivers: opportunities and limitations of target-based classification of tumors and the role of Molecular Tumor Boards. *ESMO Open* 6:100040.

Dang CV, Reddy EP, Shokat KM, Soucek L. 2017. Drugging the “undruggable” cancer targets. *Nat Rev Cancer* 17:502–508.

Elliott K, Larsson E. 2021. Non-coding driver mutations in human cancer. *Nat Rev Cancer* 21:500–509.

Erazo-Oliveras A, Muñoz-Vega M, Mlih M, Thiriveedi V, Salinas ML, Rivera-Rodríguez JM, Kim E, Wright RC, Wang X, Landrock KK, et al. 2023. Mutant APC reshapes Wnt signaling plasma membrane nanodomains by altering cholesterol levels via oncogenic β-catenin. *Nat Commun* 14:4342.

Gerasimavicius L, Livesey BJ, Marsh JA. 2022. Loss-of-function, gain-of-function and dominant-negative mutations have profoundly different effects on protein structure. *Nat Commun* 13:3895.

Hodis E, Triglia ET, Kwon JYH, Biancalani T, Zakka LR, Parkar S, Hütter J-C, Buffoni L, Delorey TM, Phillips D, et al. 2022. Stepwise-edited, human melanoma models reveal mutations’ effect on tumor and microenvironment. *Science* 376:eabi8175.

Marutani M, Tonoki H, Tada M, Takahashi M, Kashiwazaki H, Hida Y, Hamada J, Asaka M, Moriuchi T. 1999. Dominant-negative mutations of the tumor suppressor p53 relating to early onset of glioblastoma multiforme. *Cancer Res* 59:4765–4769.

Ortmann CA, Kent DG, Nangalia J, Silber Y, Wedge DC, Grinfeld J, Baxter EJ, Massie CE, Papaemmanuil E, Menon S, et al. 2015. Effect of Mutation Order on Myeloproliferative Neoplasms. *N Engl J Med* 372:601–612.

Peña MV de la, Summanen PAM, Liukkonen M, Kronholm I. 2023. Chromatin structure influences rate and spectrum of spontaneous mutations in *Neurospora crassa*. *Genome Res.* 33:599–611.

Takeda H, Wei Z, Koso H, Rust AG, Yew CCK, Mann MB, Ward JM, Adams DJ, Copeland NG, Jenkins NA. 2015. Transposon mutagenesis identifies genes and evolutionary forces driving gastrointestinal tract tumor progression. *Nat Genet* 47:142–150.

de Vries A, Flores ER, Miranda B, Hsieh H-M, van Oostrom CThM, Sage J, Jacks T. 2002. Targeted point mutations of p53 lead to dominant-negative inhibition of wild-type p53 function. *Proceedings of the National Academy of Sciences* 99:2948–2953.

Waarts MR, Stonestrom AJ, Park YC, Levine RL. 2022. Targeting mutations in cancer. *J Clin Invest* 132:e154943.

Wu C-I, Ting C-T. 2004. Genes and speciation. *Nat Rev Genet* 5:114–122.

Zhang L, Shay JW. 2017. Multiple Roles of APC and its Therapeutic Implications in Colorectal Cancer. *JNCI: Journal of the National Cancer Institute* 109:djw332.